# CATFLOW: Co-generation of Slab-Adsorbate Systems via Flow Matching

**Minkyu Kim** [1]  **Nayoung Kim** [1]  **Honghui Kim** [1]  **Sungsoo Ahn** [1]

## Abstract

Discovering heterogeneous catalysts tailored for specific reaction intermediates remains a fundamental bottleneck in materials science. While traditional trial-and-error methods and recent generative models have shown promise, they struggle to capture the intrinsic coupling between surface geometry and adsorbate interactions. To address this limitation, we propose CATFLOW, a flow matching-based framework for de novo design and structure prediction of heterogeneous catalysts. Our model operates on a primitive cell-based factorized representation of the slab-adsorbate complex, reducing the number of learnable variables by an average of $9.2\times$ while explicitly encoding the surface orientation of the slab-adsorbate interface. Experiments on the Open Catalyst 2020 dataset demonstrate that CATFLOW significantly improves the structural fidelity of generated catalysts compared to autoregressive and sequential baselines. Further experiments show that the generated structures accurately capture the adsorption energy distributions of physically plausible interfaces and lie closer to thermodynamic local minima.

## 1. Introduction

Materials discovery underpins technological advances across diverse applications, from catalysts that accelerate chemical reactions (Schlögl, 2015) to batteries for energy storage (Goodenough & Park, 2013) and porous materials for gas capture (Furukawa et al., 2013). Among these, heterogeneous catalysis stands out as a domain where the conventional discovery pipeline remains prohibitively slow and resource-intensive.

Heterogeneous catalysts exhibit atomic-scale, facet-dependent reactivity, whereby specific crystallographic sur-

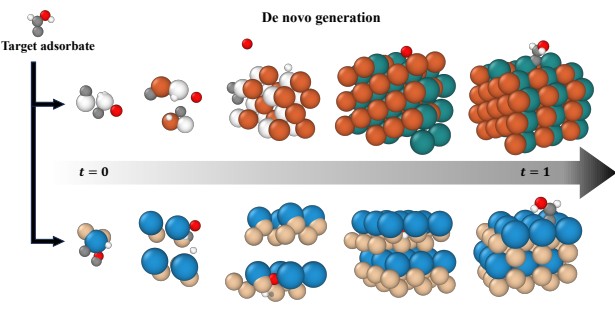

*Figure 1.* **Visualization of the co-generation trajectory conditioned on the adsorbate.** We illustrate the synchronized evolution of the slab-adsorbate system from the initial noise distribution ($t = 0$) to the final structure ($t = 1$) for de novo generation (top) and structure prediction (bottom). The framework jointly generates the components of the factorized representation to construct the slab-adsorbate system structure for the target adsorbate. At each time step, the catalyst system is explicitly constructed from these generated factorized representation components.

faces of bulk crystals govern catalytic performance (Hammer & Nørskov, 2000; Somorjai & Li, 2010). In computational modeling, such surfaces are commonly built using slab geometries that represent periodically repeating atomic layers separated by vacuum (Sun & Ceder, 2013). Under this framework, the discovery of heterogeneous catalysts can be formulated as the identification of promising bulk crystals, optimal surface facets, and favorable adsorbate binding sites (Nørskov et al., 2014; Bell, 2003). This formulation, however, inherently leads to a combinatorial explosion of candidate structures (Ertl, 2010; Greeley, 2016).

The standard workflow proceeds through sequential stages: identifying a promising bulk crystal, enumerating surface orientations and the corresponding slab models, and determining candidate binding sites for each adsorbate of interest (Nørskov et al., 2009; Tran et al., 2016; Montoya & Persson, 2017). Each stage compounds computational cost since a single bulk crystal can yield dozens of distinct orientations, each with multiple candidate binding sites (Boes et al., 2019). This combinatorial explosion makes exhaustive evaluation impractical, even with modern density functional theory (Bell, 2003; Ertl, 2010; Ulissi et al., 2017).

Recent machine learning approaches have sought to accelerate portions of this workflow, yet each targets only a subset

[1]Korea Advanced Institute of Science and Technology (KAIST). Correspondence to: Minkyu Kim <minkyu-kim@kaist.ac.kr>.

*Proceedings of the $43^{rd}$ International Conference on Machine Learning*, Seoul, South Korea. PMLR 306, 2026. Copyright 2026 by the author(s).

of the pipeline. The majority of crystal generative models, such as DiffCSP (Jiao et al., 2023) and CDVAE (Xie et al., 2022), generate bulk structures but lack mechanisms for constructing slabs or reasoning about surfaces. AdsorbDiff (Kolluru & Kitchin, 2024) places adsorbates on fixed slab inputs through conditional generation of translation and rotation, but does not generate the underlying materials.

Methods that attempt broader coverage still face limitations. Catalyst GFlowNet (Podina et al., 2025) generates bulk crystals and Miller indices but restricts to 12 single-element compositions with up to 4 atoms per unit cell and only hydrogen as the adsorbate. CatGPT (Mok & Back, 2024) tokenizes slab-adsorbate structures into a sentence, appending adsorbate atoms after the slab; it fails to capture surface-adsorbate interactions and cannot condition on adsorbates that are not described by the existing tokens.

**Contribution.** We introduce CATFLOW, a flow matching framework that addresses the catalyst discovery pipeline in one shot by co-generating slab structures and adsorbate coordinates jointly, as illustrated in Figure 1. We instantiate this framework for two tasks. For *de novo generation*, CATFLOW co-generates catalytic system structures using discrete flow matching for atomic species and continuous flow matching for atomic coordinates and lattice parameters, enabling the discovery of novel materials. For *structure prediction*, where the composition is known but the atomic arrangement is not, accurate prediction of coordinates and lattice parameters enables rapid evaluation of candidate catalysts without expensive density functional theory (DFT) initialization trials, which is a common bottleneck when screening pre-selected compositions from experimental or computational databases.

We further propose a *factorized representation* that exploits the redundancy inherent in slab structures constructed by replicating primitive cells. By decomposing the slab-adsorbate system into a primitive cell, transformation matrix, vacuum scaling factor, and adsorbate, we reduce modeling dimensionality by an average of $9.2\times$ (up to $96\times$) while preserving surface orientation information.

Our contributions are as follows:

- **Co-generation of slab-adsorbate systems.** We present the first framework to co-generate slab structures and adsorbate coordinates within a unified flow matching objective, directly capturing surface-adsorbate interactions.

- **Factorized representation of slab-adsorbate systems.** We decompose the slab-adsorbate system into primitive cells, transformation matrices, vacuum scaling factors, and adsorbate, reducing dimensionality while preserving the surface orientation information.

- **Empirical validation.** On the Open Catalyst 2020 benchmark (OC20; Chanussot et al., 2021), CATFLOW outperforms CatGPT on de novo generation and surpasses the two-step DiffCSP+AdsorbDiff pipeline on structure prediction.

## 2. Related Work

### 2.1. Bulk Crystal Generative Models

Generative models for inorganic crystals have achieved remarkable progress in materials discovery. CDVAE (Xie et al., 2022) employs variational autoencoders with equivariant graph neural networks to generate stable crystal structures in a learned latent space. DiffCSP (Jiao et al., 2023) applies diffusion models to jointly generate lattice parameters and fractional atomic coordinates. FlowMM (Miller et al., 2024) uses flow matching on Riemannian manifolds to respect the geometric constraints of periodic lattices. MatterGen (Zeni et al., 2025) scales diffusion models to handle diverse material types through a unified framework. Most recently, OMatG (Höllmer et al., 2025) combines discrete flow matching for atomic species with continuous flow for geometric properties.

However, these models focus exclusively on bulk crystal generation and lack mechanisms for explicit slab construction. They do not model surface interactions or atomic arrangements, which are the primary components of catalytic systems. Our framework addresses this limitation by explicitly generating slab structures to model the surface interactions with adsorbates.

### 2.2. Machine Learning for Heterogeneous Catalysts

Recently, there has been an explosion of interest in machine learning for heterogeneous catalysts, with new datasets computed from DFT calculations (Chanussot et al., 2021; Tran et al., 2023; Sahoo et al., 2025). We summarize the differences in Table 1.

Catalyst GFlowNet (Podina et al., 2025) generates bulk compositions and Miller indices specifying surface orientations, using pretrained graph neural network (GNN) models as proxy rewards to guide sampling toward low energy regions, but restricts the chemical space to 12 single-element compositions with up to 4 atoms per bulk unit cell. It does not explicitly incorporate adsorbate placement during the generation process. It also evaluates adsorption energy from disconnected slab-adsorbate graphs without explicit adsorbate placement. These constraints prevent exploration of diverse slab structures and modeling of slab-adsorbate interactions.

AdsorbDiff (Kolluru & Kitchin, 2024) uses diffusion models to predict adsorbate binding configurations by generating

*Table 1.* **Comparison of generative capabilities.** We contrast CATFLOW with prior work across five key functional aspects: slab generation (*Slab gen*), structure prediction from composition (*Struct pred*), adsorbate positioning (*Ads pos*), conditioning on adsorbates (*Ads cond*), and joint co-generation (*Co-gen*). CATFLOW is the only framework that satisfies all criteria, enabling end-to-end generation of catalyst-adsorbate systems.

| Method | Slab gen | Struct pred | Ads pos | Ads cond | Co-gen |
|---|---|---|---|---|---|
| Catalyst GFN | ▲ | ✗ | ✗ | ✗ | ✗ |
| AdsorbDiff | ✗ | ✗ | ✔ | ✗ | ✗ |
| CatGPT | ✔ | ✗ | ✔ | ✔ | ✗ |
| **CATFLOW** | ✔ | ✔ | ✔ | ✔ | ✔ |

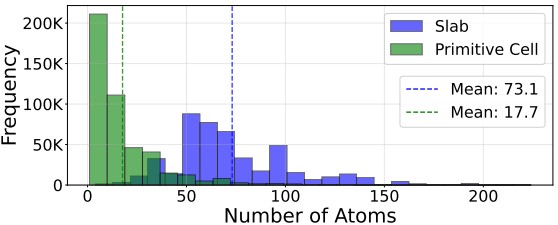

*Figure 2.* **Histogram of atom counts in catalyst structures.** We compare the histograms of atom counts for slab structures (blue) and their corresponding primitive cells (green). The primitive cells require fewer atoms than the slab structures, reducing the number of learnable variables for the generative model.

translations and rotations of adsorbate molecules, training on relaxed structures from OC20-dense trajectories (Lan et al., 2023). While AdsorbDiff effectively learns local binding geometries, it requires the slab structure as fixed input and cannot discover novel surface configurations. This framework overlooks slab generation, which is an essential step for exploring diverse and favorable surface configurations for given adsorbates.

CatGPT (Mok & Back, 2024) tokenizes slab-adsorbate structures into strings and generates them autoregressively. However, it generates slab atoms first and appends adsorbate atoms sequentially, treating the two components independently rather than modeling their interaction. Furthermore, its WordLevel tokenizer requires exact adsorbate symbol strings as input prompts for conditional generation, restricting the model to a closed set of predefined symbols and hindering generalization to arbitrary adsorbate types. In contrast, our framework co-generates slab structures and adsorbate positions within a unified flow matching objective. By conditioning on atomic species rather than predefined symbols, our framework is designed to handle arbitrary adsorbate compositions beyond the training data.

### 2.3. Machine Learning for Molecular Complexes

Following the success of AlphaFold2 (Jumper et al., 2021) in predicting single-protein structures, a natural next step has been to model complex interactions between multiple molecular entities. In the biomolecular domain, AlphaFold3 (Abramson et al., 2024), BoltzGen (Stark et al., 2025), and FlexDock (Corso et al., 2025) exemplify this direction by jointly predicting protein–ligand and protein–nucleic acid complexes. Machine learning for heterogeneous catalysis and metal-organic frameworks (Kim et al., 2026) similarly models host–guest interactions.

## 3. Method

### 3.1. Factorized Slab-Adsorbate Representation

We first describe the structure of slab-adsorbate systems and then introduce our factorized representation that decom-

poses the structure into four components: primitive cell, transformation matrix, vacuum scaling factor, and adsorbate configurations. Our factorized representation is designed to resolve redundancies in the slab-adsorbate system, reducing the number of atoms as shown in Figure 2 for the Open Catalyst 2020 (Chanussot et al., 2021) dataset. This reduction lowers the number of learnable variables that scale with the number of atoms, enabling efficient training and inference for the generative model. It also explicitly describes the surface orientation (extracted from the transformation matrix) required for interpreting catalytic systems.

At a high level, our framework decomposes the slab-adsorbate system into four learnable components: primitive cell, transformation matrix, vacuum scaling factor, and adsorbate. Figure 3 illustrates the construction of the slab-adsorbate system by the factorized representation.

**Primitive cell** $\mathcal{S}_{\text{prim}} = (\boldsymbol{a}_{\text{prim}}, \boldsymbol{x}_{\text{prim}}, \boldsymbol{L}_{\text{prim}})$ is the repeating unit of the slab, where $\boldsymbol{a}_{\text{prim}} \in \mathcal{A}^N$ denotes atomic species from the element set $\mathcal{A}$, $\boldsymbol{x}_{\text{prim}} \in \mathbb{R}^{N \times 3}$ denotes coordinates of the $N$ atoms, and $\boldsymbol{L}_{\text{prim}} \in \mathbb{R}^{3 \times 3}$ is the lattice matrix that determines the crystal periodicity.

**Transformation matrix** $M \in \mathbb{Z}^{3 \times 3}$ specifies how to construct the slab from the primitive cell. Each row represents a slab lattice vector as a linear combination of the primitive lattice vectors, giving $\boldsymbol{L}_{\text{slab}} = \boldsymbol{M} \boldsymbol{L}_{\text{prim}}$. The slab structure is generated by replicating the primitive cell atoms at all translation vectors, defined as integer linear combinations of the primitive lattice vectors, that lie within the slab unit cell. Consequently, the slab contains $|\det(\boldsymbol{M})|$ times as many atoms as the primitive cell.

**Vacuum scaling factor** $k_{\text{vac}} \in \mathbb{R}_{>1}$ determines the total height of the slab-adsorbate system cell relative to the slab thickness. Let the slab thickness be $d_{\text{slab}}$ along the $c$-axis, the system cell has $c$-vector with the length of $k_{\text{vac}} \cdot d_{\text{slab}}$, creating a vacuum region of height $(k_{\text{vac}} - 1) \cdot d_{\text{slab}}$. The system lattice is $\boldsymbol{L}_{\text{sys}} = \text{diag}(1, 1, k_{\text{vac}}) \boldsymbol{L}_{\text{slab}}$.

**Adsorbate** consists of the atomic species $\boldsymbol{a}_{\text{ads}}$ and the Cartesian atomic coordinates $\boldsymbol{x}_{\text{ads}} \in \mathbb{R}^{M \times 3}$, sharing the system

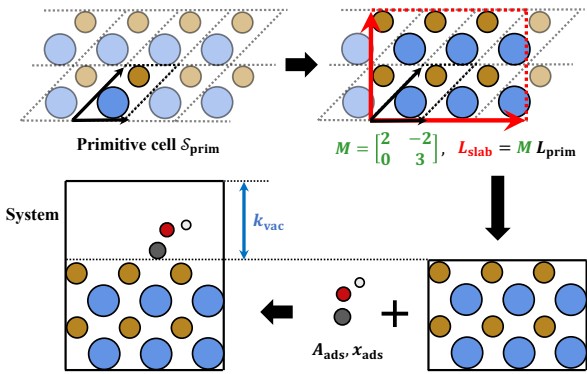

*Figure 3.* **Conceptual description of factorized representation.** The primitive cell $\mathcal{S}_{\text{prim}}$ (top left) is transformed by the transformation matrix $M$ to construct the slab lattice $L_{\text{slab}} = M\,L_{\text{prim}}$ (top right). The slab structure is generated by replicating primitive cell atoms at all translation vectors lying within $L_{\text{slab}}$ (bottom right). The vacuum scaling factor $k_{\text{vac}}$ extends $L_{\text{slab}}$ along the vertical axis to create the system cell, and the adsorbate atoms defined by the atomic species $A_{\text{ads}}$ and the atomic coordinates $X_{\text{ads}}$ are placed on the slab surface to form the complete slab-adsorbate system (bottom left).

cell $L_{\text{sys}}$ with the slab. In our framework, we treat the atomic coordinates of the adsorbate $x_{\text{ads}}$ as learnable components, while the atomic species of the adsorbate $a_{\text{ads}}$ serves as a condition.

### 3.2. Co-Generation via Flow Matching

We first describe our framework for de novo generation, then extend the framework to structure prediction. Our goal is to learn the conditional joint distribution $p(\mathcal{S}_{\text{prim}}, M, k_{\text{vac}}, x_{\text{ads}} \mid a_{\text{ads}})$, where $a_{\text{ads}}$ are the atomic species of the adsorbate.

We use continuous and discrete flow matching objectives to train a single neural network that co-generates the slab-adsorbate complex. We partition the learnable variables into continuous geometric variables $z$ and discrete compositional variables $a_{\text{prim}}$.

The continuous variables are defined as $z = (x_{\text{prim}}, \ell, M, k_{\text{vac}}, \bar{x}_{\text{ads}}, \hat{x}_{\text{ads}})$, where $\ell$ and $(\bar{x}_{\text{ads}}, \hat{x}_{\text{ads}})$ are reparameterizations of the lattice parameter $L$ and adsorbate coordinates $x_{\text{ads}}$, respectively. To be specific, $\ell = (a, b, c, \alpha, \beta, \gamma) \in \mathbb{R}^6$ denotes the Niggli-reduced lattice parameters, $\bar{x}_{\text{ads}} \in \mathbb{R}^3$ denotes the center of mass of the adsorbate, and $\hat{x}_{\text{ads}} \in \mathbb{R}^{M \times 3}$ denotes the relative positions of adsorbate atoms with respect to $\bar{x}_{\text{ads}}$.

Finally, note that we relax the integer-valued transformation matrix $M$ to continuous values during training. This relaxation is physically meaningful: a real-valued $M$ still defines a valid surface orientation via $L_{\text{slab}} = M L_{\text{prim}}$, though the resulting slab may not be a perfect replication of the

primitive cell. The decomposition of adsorbate coordinates into $\bar{x}_{\text{ads}}$ and $\hat{x}_{\text{ads}}$ decouples where the adsorbate is placed from how its atoms are arranged, allowing the model to learn placement and internal geometry separately.

**Probability path.** Following the conditional flow matching framework (Lipman et al., 2023), we define the probability path for $z_t$ via linear interpolation between the prior samples $z_0$ and the data samples $z_1$:

$$z_t = (1-t)z_0 + tz_1, \quad \text{where } z_0 \sim p_0,\ z_1 \sim q, \quad (1)$$

where $t \in [0, 1]$, $p_0$ is the prior distribution and $q$ is the data distribution. This interpolation applies component-wise to the concatenated geometric variables.

For discrete variables, we use discrete flow matching with a masking mechanism (Höllmer et al., 2025). We introduce a mask token, extending the atomic vocabulary to $\{[\texttt{MASK}]\} \cup \mathcal{A}$. The probability path interpolates from a fully masked state $a_0$ (all atoms set to $[\texttt{MASK}]$) at $t = 0$ to the true composition $a_1 = a_{\text{prim}}$ at $t = 1$. For an atomic species $a \in \{[\texttt{MASK}]\} \cup \mathcal{A}$ of a single atom within $a_1$, the probability path is defined as follows:

$$p_t(a_t \mid a_1) = (1-t)\delta_{a_t, [\texttt{MASK}]} + t\delta_{a_t, a_1}, \quad (2)$$

where $\delta_{i,j}$ denotes the Kronecker delta function, which equals 1 if $i = j$ and 0 otherwise. Thus, at time $t$, the atom remains masked with probability $(1-t)$ and reveals its true species $a_1$ with probability $t$.

We sample the primitive cell atomic coordinates $x_{\text{prim}}$ and the relative positions of the adsorbate $\hat{x}_{\text{ads}}$ from a Gaussian distribution $\mathcal{N}(\mu_{\text{coord}}, \sigma_{\text{coord}}^2 I)$, where $\mu_{\text{coord}}$ and $\sigma_{\text{coord}}$ are the empirical statistics from the training set. Similarly, the vacuum scaling factor $k_{\text{vac}}$ and the center of mass of the adsorbate $\bar{x}_{\text{ads}}$ are also sampled from a Gaussian distribution. For the transformation matrix $M$, we introduce a perturbation to the identity matrix $M = I + \epsilon$ where $\epsilon \sim \mathcal{N}(0, I)$. For the lattice parameters, we sample the lengths $a, b, c$ independently from log-normal distributions defined as $\text{LogNormal}(\mu_{\text{lat}}, \sigma_{\text{lat}}^2)$, where parameters $\mu_{\text{lat}}$ and $\sigma_{\text{lat}}$ are empirical statistics, and the lattice angles $\alpha, \beta, \gamma$ are sampled uniformly from the range $[60°, 120°]$.

**Conditional vector field.** The conditional vector field $u_t$ generating the probability path is defined as:

$$u_t(z \mid z_1, z_0) = z_1 - z_0. \quad (3)$$

It follows that the marginal vector field points toward the conditional expectation of the data (Eijkelboom et al., 2024). We parameterize our continuous decoder $\hat{z}_\theta(a_t, z_t, t, a_{\text{ads}})$ to predict the conditional expectation of data.

For discrete compositional variables, we train a classifier $p_\theta(a_1 \mid a_t, z_t, t, a_{\text{ads}})$ to predict the true composition from

the partially masked state. The classifier outputs logits over atomic species for each atom in the primitive cell.

**Loss functions.** The total loss is a linear combination of the continuous and discrete flow matching losses:

$$\mathcal{L}(\theta) = \mathcal{L}_{\text{CFM}}(\theta) + \mathcal{L}_{\text{DFM}}(\theta). \quad (4)$$

We employ the $L_1$ norm for the continuous variables, as it has been shown to effectively capture geometric structures in flow matching frameworks (Zaghen et al., 2025):

$$\mathcal{L}_{\text{CFM}}(\theta) = \mathbb{E}_{t, \boldsymbol{z}_0, \boldsymbol{z}_1, \boldsymbol{a}_t} \left[ \sum_i \lambda_i \left\| \hat{\boldsymbol{z}}_\theta^{(i)} - \boldsymbol{z}_1^{(i)} \right\|_1 \right], \quad (5)$$

where $t \sim \mathcal{U}[0, 1]$, and $i$ indexes the components of $\boldsymbol{z}$, e.g., $\boldsymbol{z}_1^{(i)}$ being the target primitive cell coordinates or the lattice parameter. Here, $\lambda_i$ denotes the loss weight for each component, and the decoder prediction is $\hat{\boldsymbol{z}}_\theta^{(i)} = \hat{\boldsymbol{z}}_\theta^{(i)}(\boldsymbol{a}_t, \boldsymbol{z}_t, t, \boldsymbol{a}_{\text{ads}})$. The discrete flow matching loss for the atomic species of the primitive cell is formulated as:

$$\mathcal{L}_{\text{DFM}}(\theta) = \mathbb{E}_{t, \boldsymbol{z}_t, \boldsymbol{a}_1, \boldsymbol{a}_t} \left[ -\log p_\theta(\boldsymbol{a}_1 \mid \boldsymbol{a}_t, \boldsymbol{z}_t, t, \boldsymbol{a}_{\text{ads}}) \right]. \quad (6)$$

**Inference.** We jointly generate the continuous and discrete components during inference. For continuous variables, we solve the ordinary differential equation from $t = 0$ to $t = 1$ using a numerical ODE solver. The vector field is derived as $(\hat{\boldsymbol{z}}_\theta - \boldsymbol{z}_t)/(1 - t)$. At $t = 1$, we round the generated transformation matrix to integers to enforce the integer constraint required for a valid supercell.

For discrete variables, we perform iterative unmasking. At each timestep $t$, we predict a candidate clean composition $\hat{\boldsymbol{a}}_1$ via categorical sampling from the model logits. We then determine whether to reveal each remaining masked token by performing a Bernoulli trial with probability $r = \Delta t/(1 - t)$. Tokens selected for unmasking are updated to the values sampled from $\hat{\boldsymbol{a}}_1$, while previously revealed tokens remain fixed. This schedule guarantees that the unmasking probability equals one at the final timestep.

**From de novo generation to structure prediction.** Structure prediction addresses scenarios where the elemental composition is already determined, e.g., from high-throughput screening, experimental synthesis constraints, or domain expertise, but the atomic arrangement remains unknown. In this setting, we are given the atomic species of the primitive cell $\boldsymbol{a}_{\text{prim}}$ and the adsorbate species $\boldsymbol{a}_{\text{ads}}$, and the goal is to predict the geometric variables: atomic coordinates $\boldsymbol{x}_{\text{prim}}$, lattice parameters $\boldsymbol{\ell}$, transformation matrix $\boldsymbol{M}$, vacuum scaling factor $k_{\text{vac}}$, and adsorbate coordinates $\boldsymbol{x}_{\text{ads}}$.

We adapt our framework by fixing the discrete compositional variables and learning only the continuous flow.

Specifically, the atomic species $\boldsymbol{a}_{\text{prim}}$ is directly embedded without masking and serves as an additional condition alongside $\boldsymbol{a}_{\text{ads}}$. The model is trained using only the continuous flow matching loss in Equation (5) with the decoder $\hat{\boldsymbol{z}}_\theta(\boldsymbol{z}_t, t, \boldsymbol{a}_{\text{prim}}, \boldsymbol{a}_{\text{ads}})$. At the inference step, we solve the ODE from $t = 0$ to $t = 1$ to generate the geometric variables, without the iterative unmasking procedure required for de novo generation.

This formulation naturally extends our co-generation framework: the same architecture captures surface-adsorbate interactions, but with composition fixed, the model focuses entirely on learning the geometric distribution conditioned on the given atomic species.

### 3.3. Neural Network Architecture

We parameterize the flow matching model using a transformer-based architecture that jointly processes primitive cell atoms and adsorbate atoms. The model follows a hierarchy of an encoder that constructs atom-level representations, a token transformer that captures global interactions, and a decoder that produces predictions for each component of the factorized representation.

**Input representation.** The encoder constructs atom-level features by combining element embeddings with geometric encodings. For primitive cell atoms, we embed atomic elements as one-hot vectors (with an additional mask token for de novo generation). For adsorbate atoms, we additionally embed reference atomic coordinates constructed using a deterministic algorithm based on molecular connectivity and covalent radii (Chanussot et al., 2021). Geometric information, including noisy primitive cell coordinates $\boldsymbol{x}_{\text{prim},t}$, noisy adsorbate coordinates $\boldsymbol{x}_{\text{ads},t} = \bar{x}_{\text{ads,t}} + \hat{\boldsymbol{x}}_{\text{ads,t}}$, lattice parameters $\boldsymbol{L}_{\text{prim},t}$, transformation matrix $\boldsymbol{M}_t$, vacuum scaling factor $k_{\text{vac},t}$, and the supercell lattice $\boldsymbol{M}_t \boldsymbol{L}_{\text{prim},t}$, are projected and incorporated into atom features.

**Encoder and token transformer.** The encoder applies self-attention across all atoms using diffusion transformer (DiT; Peebles & Xie, 2023) blocks with adaptive layer normalization (Xu et al., 2019) conditioned on timestep embeddings. These atom-level features are then processed by the token transformer, which applies additional DiT blocks to refine the representations and capture complex interactions between the primitive slab and adsorbate while maintaining the atom-level granularity.

**Decoder.** The decoder processes the joint atom representations through DiT blocks and produces predictions via separate output heads. Atom-level properties, such as atomic coordinates of primitive cell $\boldsymbol{x}_{\text{prim}}$ and adsorbate relative positions $\hat{\boldsymbol{x}}_{\text{ads}}$, are predicted for each individual atom. In contrast, system-level properties, including the adsorbate

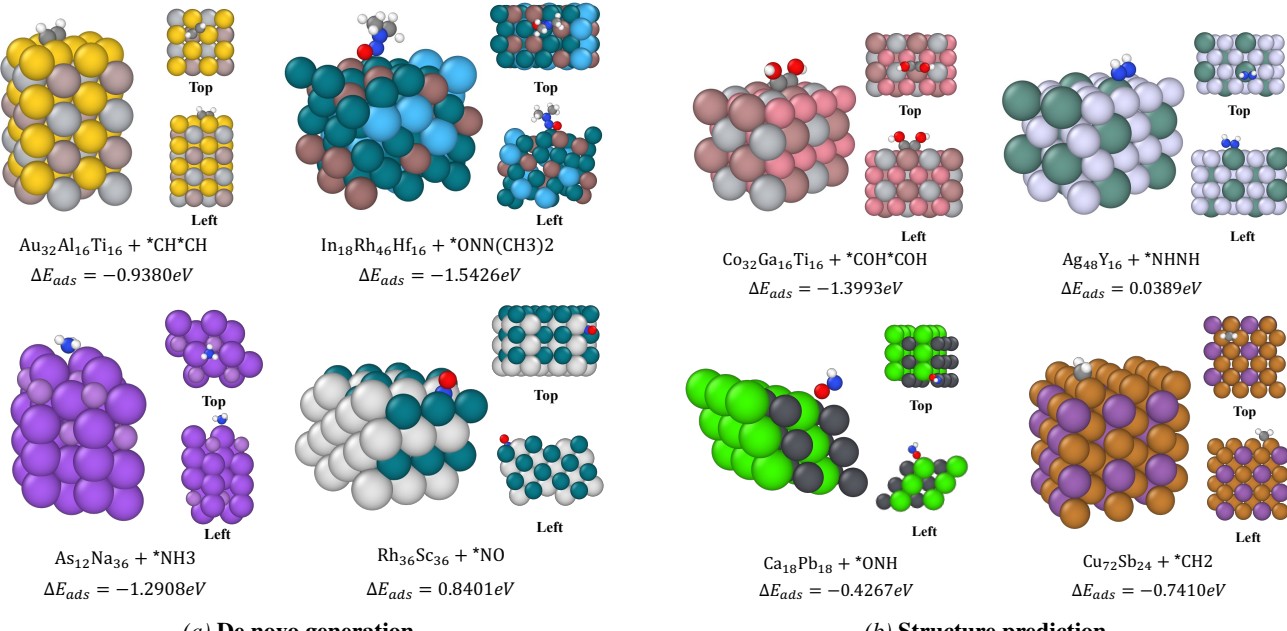

$Au_{32}Al_{16}Ti_{16}$ + *CH*CH
$\Delta E_{ads} = -0.9380 eV$

$In_{18}Rh_{46}Hf_{16}$ + *ONN(CH3)2
$\Delta E_{ads} = -1.5426 eV$

$Co_{32}Ga_{16}Ti_{16}$ + *COH*COH
$\Delta E_{ads} = -1.3993 eV$

$Ag_{48}Y_{16}$ + *NHNH
$\Delta E_{ads} = 0.0389 eV$

$As_{12}Na_{36}$ + *NH3
$\Delta E_{ads} = -1.2908 eV$

$Rh_{36}Sc_{36}$ + *NO
$\Delta E_{ads} = 0.8401 eV$

$Ca_{18}Pb_{18}$ + *ONH
$\Delta E_{ads} = -0.4267 eV$

$Cu_{72}Sb_{24}$ + *CH2
$\Delta E_{ads} = -0.7410 eV$

*(a)* **De novo generation**      *(b)* **Structure prediction**

*Figure 4.* **Visualization of generated slab-adsorbate structures.** We present generated samples for (a) de novo generation and (b) structure prediction tasks. The multi-view renderings (perspective, top, and left) illustrate that CATFLOW constructs geometrically precise structures capable of accommodating diverse and bulky adsorbates. The accompanying adsorption energies further confirm that these generated configurations are physically reasonable and situated in stable local minima.

center $\bar{x}_{ads}$, lattice parameters $\boldsymbol{L}_{prim}$, transformation matrix $\boldsymbol{M}$, and vacuum scaling factor $k_{vac}$, are derived by aggregating relevant features through a pooling operation.

Further implementation details regarding the model architecture, including specific hyperparameters as well as the complete training configurations, are provided in Appendix A.

## 4. Experiments

### 4.1. Experimental Setup

**Data processing.** We utilize the OC20 IS2RES dataset (Chanussot et al., 2021), which contains initial and DFT-relaxed structures of slab-adsorbate systems. To ensure our model can represent the slab via a primitive cell, we reconstruct the initial slab structures using the `fairchem-core` library. This is a critical step because relaxed surfaces often contain atomic perturbations that break the periodicity for primitive cell decomposition.

We process the data by extracting structural metadata, including the source bulk Materials Project ID, Miller indices, the vertical shift of the surface, and top or bottom surface selection. By supplying these parameters to the `SlabGenerator` in `pymatgen`, we reconstruct the unique slab configuration and identify its vertical dimensions, such as the number of slab and vacuum layers, to compute the vacuum scaling factor $k_{vac}$. The reconstructed slab is then decomposed into its primitive lattice $\boldsymbol{L}_{prim}$ and

structure $\mathcal{S}_{prim}$ using the `get_primitive_structure` function provided in `pymatgen`, with the transformation matrix $\boldsymbol{M} = \boldsymbol{L}_{slab}\boldsymbol{L}_{prim}^{-1}$.

Finally, we incorporate the adsorbate coordinates $\boldsymbol{x}_{ads}$ obtained from the relaxed structures. By co-generating the system with the adsorbate already situated in an adsorption-favorable configuration, we significantly narrow the search space for subsequent structural relaxation, facilitating convergence toward stable binding sites. This strategy shares the same motivation as AdsorbDiff (Kolluru & Kitchin, 2024), which employs diffusion models to predict favorable initial adsorbate placements before relaxation.

We extract the primitive cell $\mathcal{S}_{prim}$, the transformation matrix $\boldsymbol{M}$, and the vacuum scaling factor $k_{vac}$ from the initial structures to ensure consistent primitive cell decomposition. And we source the atomic coordinates of the adsorbate $\boldsymbol{x}_{ads}$ directly from the relaxed structures. This strategy enables the model to predict energetically favorable binding configurations (Kolluru & Kitchin, 2024). Detailed data processing procedures are provided in Appendix B.

**Baselines.** For de novo generation, we compare against CatGPT (Mok & Back, 2024), an autoregressive model that generates catalyst components via conditional text generation, i.e., lattice parameters, the atomic numbers and the atomic coordinates of bulk, surface, and adsorbate atoms.

For structure prediction, no existing framework addresses

*Table 2.* **Performance comparison on de novo generation task.** We evaluate the generative quality of CATFLOW against the monolithic baseline (CatGPT) using fundamental metrics for structural validity, uniqueness, and compositional diversity. We further analyze relaxation statistics, including system energy ($\Delta E_{sys}$), convergence steps, and success rates to verify the thermodynamic stability and optimization efficiency of the generated samples. CATFLOW outperforms the baseline across all metrics, demonstrating its ability to generate diverse structures that are not only geometrically valid but also positioned closer to their local minima of adsorption energy. **Bold** indicates the best performance.

| Method | Validity (%) ↑ | Uniqueness (%) ↑ | Comp. diversity ↑ | $\Delta E_{sys}$ (eV) ↓ | Conv. steps ↓ | Conv. rate (%) ↑ |
|---|---|---|---|---|---|---|
| CatGPT | 92.67 | 79.91 | 15.0655 | 33.1267 | 121.0629 | 98.26 |
| CATFLOW | **97.33** | **94.69** | **15.0724** | **28.0060** | **115.0534** | **99.22** |

the task. So we establish a comparative baseline by constructing a two-step modular pipeline that integrates DiffCSP (Jiao et al., 2023) and AdsorbDiff (Kolluru & Kitchin, 2024). In this setup, DiffCSP generates the primitive cell $\mathcal{S}_{\text{prim}}$ by predicting the lattice matrices and fractional coordinates for given atomic species of primitive cell $\boldsymbol{A}_{\text{prim}}$ via diffusion. After we expand the primitive cell into a slab using the ground-truth transformation $M$ and vacuum scaling factor $k_{\text{vac}}$, AdsorbDiff positions the adsorbate configuration on the slab structure. We note that this comparison favors the baseline: the two-step pipeline has access to the ground-truth transformation matrix and vacuum scaling factor, while CATFLOW predict these quantities from scratch.

**Evaluation protocol.** The primary metric characterizing interactions is the adsorption energy $\Delta E_{\text{ads}}$, defined as:

$$\Delta E_{\text{ads}} = E_{\text{sys}} - E_{\text{slab}} - E_{\text{ads}} . \tag{7}$$

To compute the energy of the slab-adsorbate system $E_{\text{sys}}$ and the energy of the slab $E_{\text{slab}}$, we perform structural relaxation using the L-BFGS algorithm with the OC20-pretrained `uma-s-1p1` model (Wood et al., 2025). This optimization protocol is applied to both our generated samples and the reference samples from the OC20 validation set in-distribution (Val ID) to establish reference adsorption energies $\Delta E_{\text{ads}}^{\text{ref}}$. The energy of the adsorbate $E_{\text{ads}}$ is derived from a linear combination of potential energies of the reference molecules $N_2$, $H_2O$, $CO$, and $H_2$ (Chanussot et al., 2021) calculated by the same model. All relaxations utilize a force convergence threshold of $f\text{max} = 0.05 \text{ eV/Å}$ with a maximum of 500 steps, and non-converged samples are excluded from the analysis.

In both de novo generation and structure prediction, we generate the samples using 50 integration steps. Structural validity assesses whether the generated structures satisfy physical constraints: all interatomic distances exceeding $0.5 \text{ Å}$ and the cell volume being greater than $0.1 \text{ Å}^3$.

For the de novo generation, we evaluate uniqueness and compositional diversity. Uniqueness measures the percentage of non-duplicate structures within the generated slab structures as determined by the `StructureMatcher` in `pymatgen`. Compositional diversity is quantified by the

average pairwise Euclidean distance between the compositional fingerprints. We also analyze convergence statistics during relaxation to assess the proximity of the model-generated initial structures to their local equilibrium states.

For structure prediction, we evaluate the match rate (MR) and root mean square deviation (RMSD) between the generated slab structures and the ground truths, generating one sample per given composition. We employ `StructureMatcher` to compute these metrics with tolerances of `ltol = 0.3` for fractional lengths, `stol = 0.5` for sites, and `angle_tol = 10` degrees, consistent with standard crystal structure prediction benchmarks (Xie et al., 2022; Jiao et al., 2023; Miller et al., 2024; Höllmer et al., 2025). For assessing the prediction quality for $\Delta E_{\text{ads}}$, we follow the protocol of Kolluru & Kitchin (2024) with a success criterion of $|\Delta E_{\text{ads}}^{\text{ref}} - \Delta E_{\text{ads}}^{\text{gen}}| \leq 0.1 \text{ eV}$.

### 4.2. Results

**De novo generation.** CATFLOW outperforms the baseline (CatGPT) across all metrics in Table 2. The higher structural validity demonstrates that our model generates more physically plausible structures for a given number of sampling attempts. The superior uniqueness and compositional diversity scores indicate that CATFLOW explores the chemical space more effectively, producing novel and diverse slab structures as illustrated in Figure 4. Furthermore, CATFLOW generates initial configurations closer to local minima in the adsorption energy landscape, as evidenced by the lower energy difference $\Delta E_{\text{sys}}$ between initial and relaxed slab-adsorbate systems. The generated samples also converge more frequently and require fewer optimization steps on average.

We further validate the energetic quality by analyzing the adsorption energy distributions from CATFLOW, CatGPT, and the reference samples for 67 validation adsorbates (the sample count for each adsorbate is based on its frequency in the validation set). We find that for 40 adsorbates, CATFLOW generates energy distributions closer to the reference adsorption energies $\Delta E_{\text{ads}}^{\text{ref}}$ than the baseline. We visualize the distributions using Kernel Density Estimation (KDE) plots for four representative adsorbates in Figure 5, selected

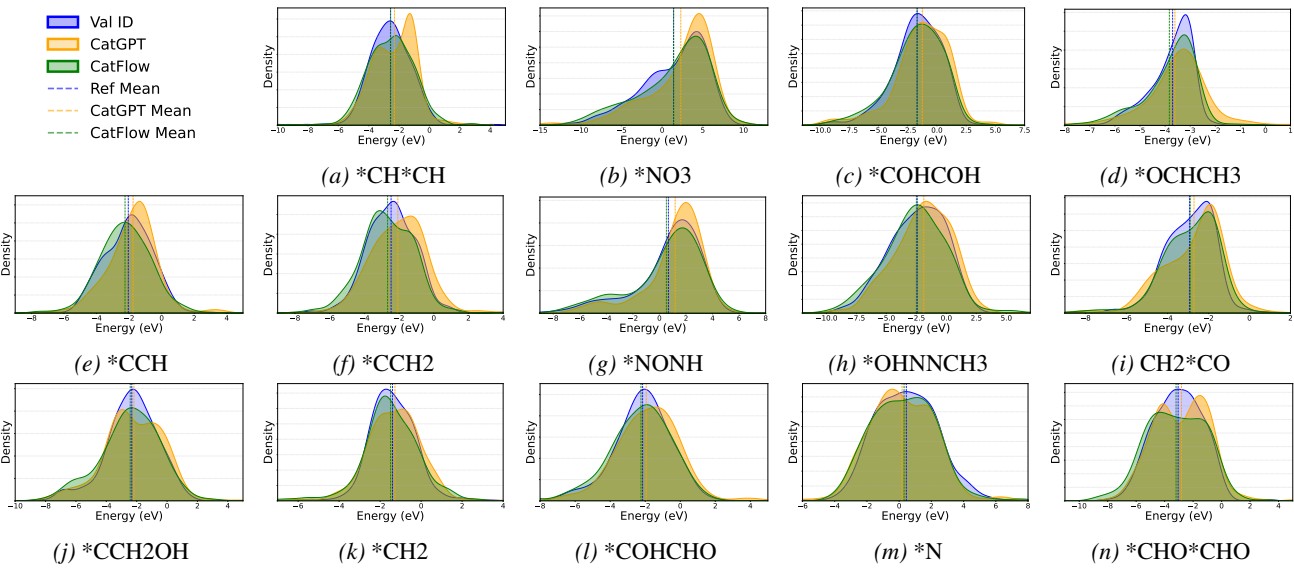

*Figure 5.* **Adsorption energy histograms in de novo generation.** Comparison of adsorption energy distributions for representative adsorbates, randomly selected to demonstrate diverse cases. Each panel shows the kernel density estimation (KDE) plots for the validation set in distribution (Val ID) (blue), CatGPT (orange), and CATFLOW (green). The vertical dashed lines indicate the mean adsorption energies for each distribution. Qualitatively, the energy profiles generated by CATFLOW exhibit stronger overlap with the validation set across most adsorbates, particularly for complex molecules such as *CH*CH, *NO3, *COHCOH, and *OCHCH3, while CatGPT shows extended tails toward positive energies, indicating generation of unstable configurations.

*Table 3.* **Performance comparison on structure prediction task.** We evaluate structural validity, fidelity using Match Rate (MR) and RMSD relative to the ground truth, alongside thermodynamic accuracy measured by the $\Delta E_{\text{ads}}$ success rate (tolerance $\leq 0.1$ eV) over matched samples. CATFLOW achieves substantial improvements over the two-step baseline combining DiffCSP and AdsorbDiff (DC+AD) in all metrics. Notably, the superior success rate highlights the capability of our end-to-end framework in identifying local energy minima. **Bold** indicates the best performance.

| Method | Validity (%) ↑ | MR (%) ↑ | RMSD ↓ | Succ.(%) ↑ |
|---|---|---|---|---|
| DC+AD | 64.95 | 7.71 | 0.1069 | 10.57 |
| CATFLOW | **98.16** | **11.09** | **0.0973** | **16.46** |

*Table 4.* **Success rate and mean absolute difference of $\Delta E_{ads}$ from the corresponding global minimum.** CATFLOW trained on OC20 IS2RES is compared against random and heuristic placements, with all candidates relaxed by `uma-s-1p1`. A sample is counted as a success if $|\Delta E_{\text{ads}}^{\text{global}} - \Delta E_{\text{ads}}^{\text{gen}}| \leq 0.1\,\text{eV}$.

| Model | Success rate (%) | $|\Delta E_{\text{ads}}^{\text{global}} - \Delta E_{\text{ads}}^{\text{gen}}|$ (eV) |
|---|---|---|
| Random+Heuristic | 10.5 | 1.9472 |
| CATFLOW | **15.2** | **0.3798** |

to show the cases of diverse binding atom types and counts. These plots qualitatively demonstrate that the adsorption energy distributions generated by CATFLOW exhibit a stronger overlap with the reference adsorption energies compared to the baseline, thereby corroborating the quantitative improvements measured by the Wasserstein distance. In contrast, CatGPT tends to generate unstable configurations, as evidenced by extended tails toward positive energies.

**Structure prediction.** Table 3 demonstrates that CATFLOW surpasses the DiffCSP+AdsorbDiff pipeline across all metrics in structure prediction. CATFLOW achieves substantial margins in structural validity and match rates compared to the baseline. The success rate for predicting the adsorption energy $\Delta E_{\text{ads}}$ between the generated samples and

the reference samples approaches nearly 10%, suggesting that CATFLOW effectively predicts slab-adsorbate system structures near the local minima of the adsorption energy landscape for given slab compositions as shown in Figure 4.

Notably, this comparison is not favorable to our method. The baseline has access to the ground-truth transformation matrix $M$ and the vacuum scaling factor $k_{\text{vac}}$, while CATFLOW predicts both quantities from scratch. Despite this significant disadvantage, CATFLOW outperforms the baseline, validating the effectiveness of our end-to-end co-generation approach. This is notable given that conditioning on $A_{\text{prim}}$ inherently yields diverse valid slabs, which inevitably affects match rates and adsorption energy evaluation.

**Global minima coverage.** CATFLOW covers the global minima of slab-adsorbate systems, despite our training data (OC20 IS2RES) typically providing a single relaxed configuration per system without global optimality guarantees.

We randomly select 100 systems from OC20-Dense (Lan et al., 2023), where the global minimum adsorption energy $\Delta E_{\text{ads}}^{\text{global}}$ is known. For each system, we generate structures with CATFLOW structure prediction model and relax them using `uma-s-1p1`, then count a success when $|\Delta E_{\text{ads}}^{\text{global}} - \Delta E_{\text{ads}}^{\text{gen}}| \leq 0.1\,\text{eV}$. As a baseline, we sample the same number of configurations per system using the random and heuristic placement and relax them with the same energy evaluator. The results shown in Table 4 indicate that CATFLOW captures meaningful thermodynamic interactions for global minima search, leveraging the broad chemical coverage of OC20.

## 5. Conclusion

We presented CATFLOW, a flow matching framework that co-generates slab structures and adsorbate positions for heterogeneous catalyst design. Our factorized representation reduces the number of learnable variables by an average of $9.2\times$ while preserving surface information. Experiments on the OC20 dataset demonstrate that CATFLOW outperforms autoregressive and two-step baselines in de novo generation and structure prediction, producing structures closer to thermodynamic local minima. CATFLOW further covers the global minima of slab-adsorbate systems, indicating that the model captures meaningful thermodynamic interactions for catalyst screening. The framework naturally extends to inverse design with target adsorption energies and multi-adsorbate systems, providing a foundation for scalable generative models in catalyst discovery.

## Acknowledgements

This work was partly supported by Institute for Information & communications Technology Planning & Evaluation(IITP) grant funded by the Korea government(MSIT) (RS-2019-II190075, Artificial Intelligence Graduate School Program(KAIST)), GRDC(Global Research Development Center) Cooperative Hub Program through the National Research Foundation of Korea(NRF) grant funded by the Ministry of Science and ICT(MSIT) (No. RS-2024-00436165), the Advanced GPU Utilization Support Program(Beta Service) funded by the Government of the Republic of Korea (Ministry of Science and ICT), the Institute of Information & Communications Technology Planning & Evaluation(IITP) grant funded by the Korea government(MSIT) (RS-2025-02304967, AI Star Fellowship(KAIST)), the "Advanced GPU Utilization Support Program" funded by the Government of the Republic of Korea (Ministry of Science and ICT), and the National Research Foundation of Korea(NRF) grant funded by the Korea government(MSIT) (RS-2025-02216257).

## Impact Statement

This work aims to accelerate the discovery of heterogeneous catalysts by introducing a generative framework that efficiently models the coupling between surface geometry and adsorbate interactions by co-generation. Heterogeneous catalysis is a cornerstone technology for addressing critical global challenges, including renewable energy storage, efficient chemical synthesis, and carbon capture. By reducing the computational cost associated with traditional trial-and-error density functional theory workflows, our approach lowers the barrier to exploring vast chemical spaces. While generative models carry inherent risks regarding the physical realizability of generated samples, our focus on structural fidelity and thermodynamic stability aims to mitigate these issues for downstream experimental verification. We believe this research primarily yields positive societal impacts by facilitating the development of materials essential for a sustainable future.

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

*Table 5.* Hyperparameters and training configuration.

| Parameter | Value |
|---|---|
| *Optimization & Infrastructure* | |
| Computing Devices | 8 GPUs |
| Optimizer | AdamW |
| Learning Rate | $1 \times 10^{-4}$ |
| Warmup Steps | 5000 |
| Global Steps | 250,000 |
| Batch Size (per device) | 128 |
| *Model Architecture* | |
| Hidden Dimension | 768 |
| FFN Expansion Ratio | 4 |
| Activation Function | GELU |
| Atom Encoder (Depth / Heads) | 8 / 12 |
| Token Transformer (Depth / Heads) | 24 / 12 |
| Atom Decoder (Depth / Heads) | 8 / 12 |

## A. Details for Training

**Code and Data Availability.** The source code is available at https://github.com/minkyu1022/CatFlow, and the processed datasets with generated samples and computed energy values are provided at https://zenodo.org/records/18382309

**Model Architecture.** Table 5 provides the detailed training configurations and model hyperparameters. We utilized the AdamW optimizer with a batch size of 128 per device on an 8-GPU infrastructure. The model incorporates GELU activation and a feed-forward expansion ratio of 4 across all transformer blocks.

We provide further details on the model components. Figure 6 illustrates the Atom Attention Encoder. The encoder embeds the noisy Cartesian coordinates of both the primitive cell and the adsorbate, combining them with global conditioning inputs such as lattice parameters and vacuum scaling factors. These inputs are aggregated into a joint latent representation and processed via DiT blocks conditioned on the timestep.

The subsequent components, the Trunk and Decoder, process this representation as follows:

- **Atom Encoder:** This module embeds discrete atomic numbers and continuous positions into a unified high-dimensional latent space. It utilizes a series of attention blocks to aggregate local chemical environments and geometric contexts before passing the features to the main transformer trunk.

- **Token Transformer:** Acting as the architectural backbone, this module consists of stacked DiT blocks that process the joint sequence of slab and adsorbate tokens. It refines latent features through global self-attention mechanisms to capture complex long-range structural dependencies without altering the sequence length.

- **Atom Attention Decoder:** This component maps the refined latent features back to the geometric target space using specialized projection heads. It independently predicts the denoised coordinates for the slab and adsorbate, while simultaneously reconstructing global parameters including lattice vectors and supercell matrices.

## B. Data Processing Details

The data processing pipeline transforms raw Open Catalyst 2020 (OC20) structures into a memory-efficient factorized representation suitable for the generative model. This process involves two primary stages: metadata extraction and structural decomposition.

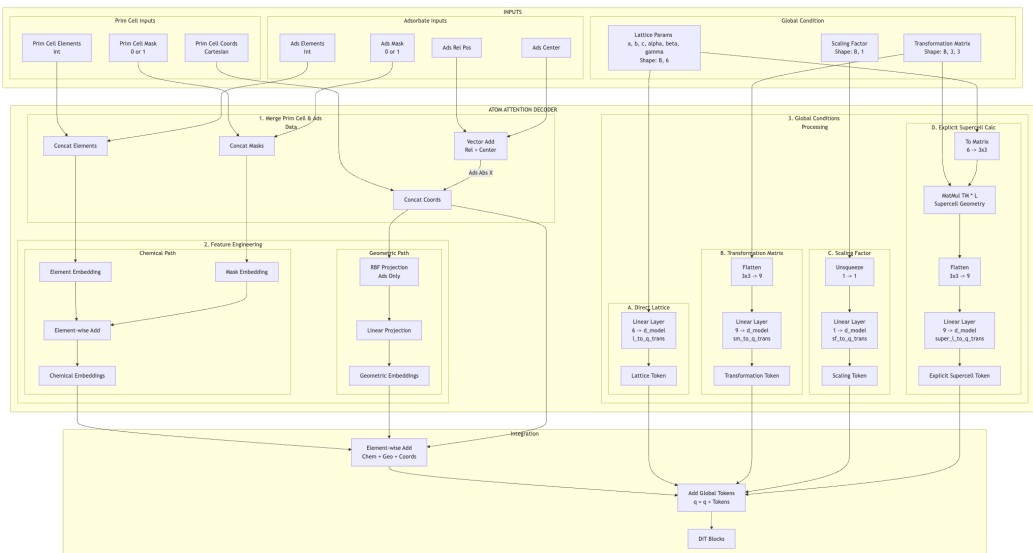

*Figure 6.* Architecture of the atom attention encoder. The encoder processes noisy coordinates of the primitive cell and adsorbate, along with global lattice parameters, to generate a joint latent representation.

### B.1. Metadata Extraction and Slab Analysis

We first extract the raw atomic structures, energies, and system identifiers from the OC20 LMDB database. To enable the decomposition of surface structures, we determine the crystallographic parameters of the slab. Using the bulk source ID (bulk_src_id) and Miller indices (miller_index) provided in the mapping files, we reconstruct the slab generation process using the Pymatgen SlabGenerator. This allows us to calculate the number of atomic layers in the slab ($n_{\text{slab}}$) and the vacuum region ($n_{\text{vac}}$), as well as the geometric height of the layers.

### B.2. Construction of Factorized Representation

Using the extracted slab parameters, we decompose the full catalyst system into four independent components: the primitive cell, the transformation matrix, the vacuum scaling factor, and the adsorbate configuration.

1. **Primitive cell extraction**: We isolate the surface atoms (tags 0 and 1) and remove the vacuum layer by compressing the unit cell along the $z$-axis based on the ratio $n_{\text{slab}}/(n_{\text{slab}} + n_{\text{vac}})$, resulting in a slab without vacuum. We then identify the primitive unit cell of this slab structure without a vacuum using a symmetry tolerance of 0.1 Å. This resulting structure is defined as the *primitive cell*.

2. **Transformation matrix calculation**: We compute the affine transformation required to map the primitive cell back to the slab as the supercell. This is represented by an integer *transformation matrix $M \in \mathbb{Z}^{3 \times 3}$*, calculated via the dot product of the slab lattice matrix and the inverse of the primitive cell lattice matrix.

3. **Vacuum scaling**: To recover the original vacuum spacing in the system lattice cell, we calculate the *vacuum scaling factor $k_{\text{vac}} = (n_{\text{slab}} + n_{\text{vac}})/n_{\text{slab}}$*. This scalar value represents the expansion required along the surface normal to restore the simulation cell dimensions.

4. **Adsorbate positioning**: The adsorbate atoms (tag 2) are separated from the surface. Their positions are standardized relative to the transformed supercell to ensure consistent alignment with the reconstructed slab, applying the translation and the rotation from the reconstructed slab structure.

### B.3. Validation and Filtering

To ensure data integrity, we perform a reconstruction test for every processed sample. We rebuild the full system using the extracted components and calculate the Root Mean Square Deviation (RMSD) between the reconstructed structure and the ground truth. We validate the structure at three stages: the tight slab, the slab with vacuum, and the full system with the

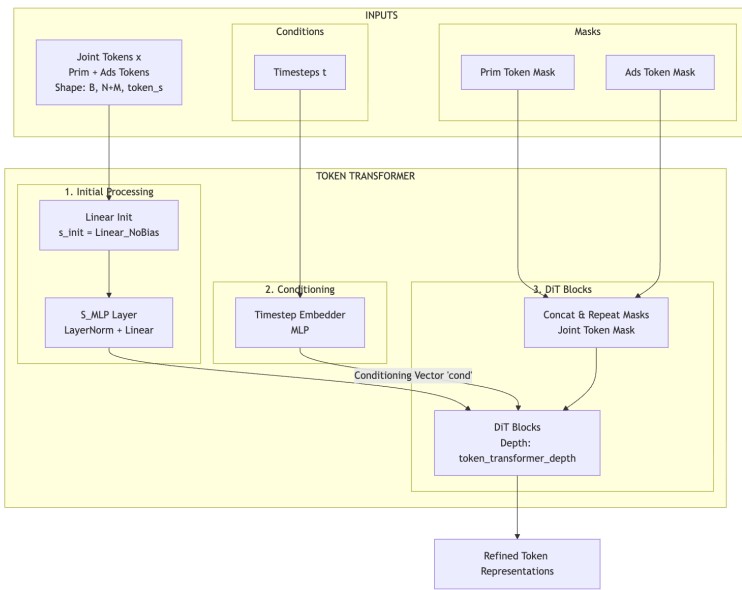

*Figure 7.* Architecture of the token transformer (trunk). This module refines the joint latent representations of primitive cell and adsorbate tokens through a series of DiT blocks, conditioned on the diffusion timestep.

adsorbate. Only samples where the RMSD is strictly less than $1 \times 10^{-5}$ for all three stages are included in the final dataset. Samples failing this criterion or raising calculation errors ($\sim 0.8\%$) are logged and excluded.

## C. Reconstruction from Factorized Representation

The reconstruction algorithm restores the slab-adsorbate system structure from the factorized representation generated by the model. This process involves a sequential pipeline that transforms the primitive cell into the lattice of the system with the adsorbate. The procedure consists of four primary steps:

1. **Primitive cell instantiation**: The primitive cell $\mathcal{S}_{\mathrm{prim}}$ is first constructed using the predicted lattice parameters $(a, b, c, \alpha, \beta, \gamma)$ and the Cartesian atomic coordinates of the primitive cell. The atomic species are assigned based on the input atomic numbers, and padding masks are applied to filter out invalid entries.

2. **Supercell transformation**: The primitive cell is expanded into the lattice of the slab as the specific surface supercell configuration. This transformation utilizes the predicted transformation matrix $M \in \mathbb{R}^{3 \times 3}$. The elements of $M$ are rounded to the nearest integers to ensure a valid periodic transformation.

3. **Vacuum addition**: To recover the lattice cell of the system required for the energy calculations with simulations, the lattice of the slab is expanded along the surface normal vector. The third unit cell vector $\mathbf{c}$ is scaled by the predicted vacuum scaling factor $k_{\mathrm{vac}}$.

4. **System integration and tagging**: The adsorbate atoms are introduced into the reconstructed cell of the system using their predicted Cartesian atomic coordinates. To facilitate downstream analysis, integer tags are assigned to all atoms: adsorbate atoms are tagged as 2, surface atoms as 1, and bulk atoms as 0. Surface atoms are identified via a height-based heuristic algorithm, selecting atoms located within 2 Å of the maximum $z$-coordinate of the slab.

## D. Experimental Details

### D.1. Uniqueness Evaluation

We evaluate the uniqueness of the generated catalyst surfaces to quantify the model's ability to explore diverse local minima without mode collapse. The uniqueness metric is computed specifically for the slab component, ensuring that variations in

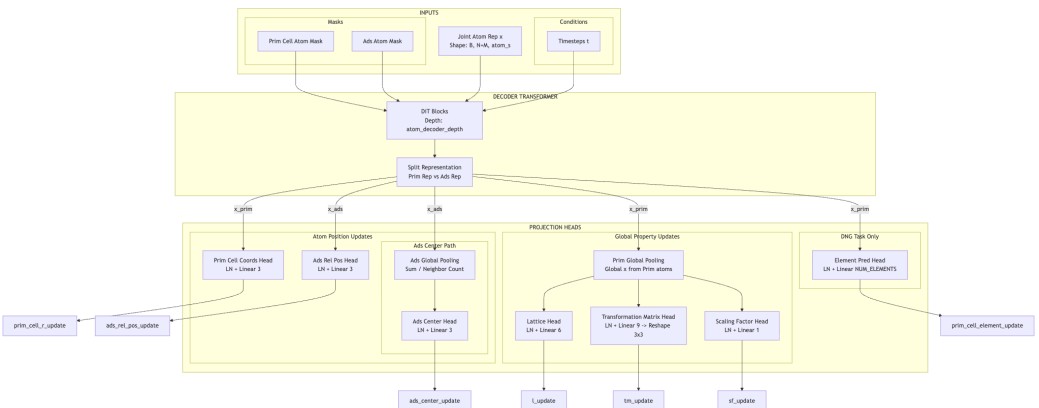

*Figure 8.* Architecture of the atom attention decoder. The decoder transforms the refined latent representations into specific updates for atom positions, lattice parameters, and the transformation matrix via specialized projection heads.

adsorbate positioning do not inflate the score. The procedure is as follows:

1. **Slab isolation:** For every generated sample, we isolate the surface structure by removing the adsorbate atoms. This is achieved by filtering atoms based on their tags, where atoms with the tag 2 (representing the adsorbate) are excluded.

2. **Structure grouping:** We employ the `StructureMatcher` class from the `pymatgen` library to identify structural duplicates. This algorithm groups structures that are equivalent under translation, rotation, and periodic boundary conditions using default tolerances ($ltol = 0.2$, $stol = 0.3$, $angle\_tol = 5$).

3. **Metric calculation:** The uniqueness score is defined as the ratio of the number of unique structure groups ($N_{\text{unique}}$) to the total number of valid generated samples ($N_{\text{total}}$):

$$\text{Uniqueness} = \frac{N_{\text{unique}}}{N_{\text{total}}} \tag{8}$$

### D.2. Compositional Diversity

To quantify the chemical variety of the generated structures, we calculate the mean pairwise distance between their compositional feature vectors. The process consists of three steps:

1. **Featurization:** We represent the chemical composition of each structure using the ElementProperty featurizer from the `matminer` library. Specifically, we employ the Magpie preset, which maps a composition to a 132-dimensional vector. This vector contains weighted statistics (mean, average deviation, range, etc.) of fundamental elemental properties such as atomic number, atomic mass, melting temperature, and electronegativity.

2. **Normalization:** To ensure that all feature dimensions contribute equally to the distance metric, we standardize the feature vectors across the entire dataset. We apply a standard scaler to transform the distribution of each feature to have a mean of 0 and a variance of 1.

3. **Metric calculation:** We define the compositional diversity $D_{\text{comp}}$ as the average Euclidean distance between the normalized feature vectors ($\mathbf{x}$) of all unique pairs in the dataset:

$$D_{\text{comp}} = \frac{2}{N(N-1)} \sum_{i<j} \|\mathbf{x}_i - \mathbf{x}_j\|_2$$

where $N$ is the total number of valid samples.

*Table 6.* **Ablation study for architecture.** We replace DiT with EGNN under the same resource budget.

| Method | Validity (%) ↑ | Uniqueness (%) ↑ | Comp. diversity ↑ |
|---|---|---|---|
| CATFLOW-EGNN | 49.68 | **95.72** | 14.8568 |
| CATFLOW-DiT | **97.33** | 94.69 | **15.0724** |

*Table 7.* **Ablation study for continuous relaxation of $M$.** We apply discrete flow matching for $M$.

| Method | Validity (%) ↑ | Uniqueness (%) ↑ | Comp. diversity ↑ |
|---|---|---|---|
| Discrete training $M$ | 81.74 | 90.89 | 15.0364 |
| CATFLOW | **97.33** | **94.69** | **15.0724** |

*Table 8.* **Ablation study for factorized representation.** We replace the factorized representation with a full-slab mixed structure.

| Method | Validity (%) ↑ | Uniqueness (%) ↑ | Comp. diversity ↑ | $\Delta E_{sys}$ (eV) ↓ | Conv. steps ↓ | Conv. rate (%) ↑ |
|---|---|---|---|---|---|---|
| Full-slab | 97.28 | **95.27** | **15.1891** | 46.8071 | 152.0243 | 97.08 |
| CATFLOW | **97.33** | 94.69 | 15.0724 | **28.0060** | **115.0534** | **99.22** |

*Table 9.* **Ablation study for co-generation.** The sequential variant first samples a primitive cell composition from the training set, then applies CATFLOW trained for structure prediction. We vary the sampling temperature ($T = 0.1, 1.0$) to control the compositional diversity. All variants are evaluated on the de novo generation task.

| Method | Validity (%) ↑ | Uniqueness (%) ↑ | Comp. diversity ↑ | $\Delta E_{sys}$ (eV) ↓ | Conv. steps ↓ | Conv. rate (%) ↑ |
|---|---|---|---|---|---|---|
| Seq. $T = 0.1$ | **98.11** | 93.05 | 13.6338 | 31.8368 | 116.9971 | 98.67 |
| Seq. $T = 1.0$ | 97.79 | 94.41 | 14.0925 | 31.7265 | 116.8787 | 98.65 |
| CATFLOW | 97.33 | **94.69** | **15.0724** | **28.0060** | **115.0534** | **99.22** |

# E. Ablation studies

We conduct ablation studies on the de novo generation task to validate four design choices of CATFLOW, namely the DiT architecture, the continuous relaxation of the transformation matrix $M$, the factorized representation, and the co-generation of slab and adsorbate. All variants share the same resource budget for a fair comparison.

The EGNN and discrete-$M$ variants exhibit large validity gaps relative to CATFLOW. Given these gaps, we prioritize structural metrics over energy-based metrics for these two variants. Energy-based metrics are time-consuming to compute and less meaningful when comparing variants with substantially different validity rates, since the relaxation statistics aggregate over only the valid structures. Uniqueness is also less interpretable under such conditions. We therefore omit the relaxation statistics for the EGNN and discrete-$M$ variants and report them only for the factorized representation and co-generation variants.

**DiT architecture.** We replace the DiT backbone with an EGNN while keeping the same resource budget. Table 6 shows that the EGNN variant drops the structural validity from 97.33% to 49.68%. This result shows that the DiT architecture is essential for generating physically plausible slab-adsorbate systems under our representation.

**Continuous relaxation of $M$.** We replace the continuous relaxation of the transformation matrix $M$ with discrete flow matching. Table 7 shows that the discrete variant lowers the structural validity from 97.33% to 81.74%. Treating $M$ as a continuous variable, therefore, yields more valid slab structures than modeling its integer entries directly.

**Factorized representation.** We replace the factorized representation with full-slab generation, where the model generates all atoms of the slab instead of a primitive cell. Table 8 shows that the full-slab variant attains comparable structural validity and compositional diversity, but increases the energy difference $\Delta E_{sys}$ from 28.0060 eV to 46.8071 eV and requires more relaxation steps on average. This result shows that the factorized representation generates initial structures closer to local minima of the adsorption energy landscape.

**Co-generation of slab and adsorbate.** We compare CATFLOW against a sequential variant that first samples a primitive cell composition from the training set, then applies CATFLOW trained for structure prediction. We vary the sampling temperature ($T = 0.1, 1.0$) to control the compositional diversity. Table 9 shows that the sequential variant reduces the compositional diversity and increases the energy difference $\Delta E_{sys}$ across both temperatures. This result shows that co-generating the slab and the adsorbate produces more diverse compositions and structures closer to local minima than generating them in sequence.

