# OpenReview forum: "CatFlow: Co-generation of Slab-Adsorbate Systems via Flow Matching"
_ICML.cc/2026/Conference — ICML 2026 regular_

### Official Review · Reviewer_r5u6 · 2026-03-12

**Soundness:** 3
**Presentation:** 3
**Significance:** 2
**Originality:** 2
**Overall Recommendation:** 4
**Confidence:** 5

**Summary:**

This paper introduces CatFlow, a hybrid flow matching approach that combines continuous and discrete domains to jointly generate slab–adsorbate configurations in heterogeneous catalysis. In order to enable such modeling, the paper introduces a factorized representation of the system into a primitive cell, supercell transformation matrix, vacuum scaling factor, and adsorbate configuration (center of mass + relative geometry). Experiments on the OC20 dataset for de novo generation and structure prediction reports improved physical validity and diversity over CatGPT and better structure-prediction metrics than a two-step DiffCSP + AdsorbDiff pipeline.

**Compliance With Llm Reviewing Policy:**

Affirmed.

**Final Justification:**

The paper proposes a technically sound framework for joint slab-adsorbate generation, with clear empirical gains, though initial concerns centered on the problem formulation and the lack of a physical stability evaluation. The rebuttal clarified that the goal is sample-efficient candidate generation rather than direct energy minimization and provided additional OC20-Dense experiments and SUN/mSUN metrics, addressing the main concerns. Originality remains moderate, as the contribution is primarily in problem formulation and representation rather than new algorithms. Overall, the rebuttal strengthened confidence in the method’s validity and practical relevance. I therefore raise my recommendation to weak accept (4).

**Key Questions For Authors:**

The questions are ordered by importance and urgency. Questions #1 and #2 are the primary issues that I will reconsider the score if the authors provide adequate responses. Please feel free to provide partial responses early so we can iterate during the discussion period.

1. Regarding the weakness of this work mentioned in the "Problem setting" point in Weaknesses (#1), could the authors modify the problem setting, dataset, and evaluation accordingly to properly capture the important physical/thermodynamic interaction between catalyst and adsorbate (that stems from the low energy configurations)? I am aware that the already published CatGPT paper includes a similar caveat by using the OC20-S2EF set, but I believe this is a sound issue and I will raise the score to >=4 given that the authors convince me that the current modeling scheme is valid or revise the work accordingly.
2. Could the authors include the standardized stability metrics for the de novo generation experiments, for example, incorporating the SUN/mSUN metrics used in crystal structure generation papers?
3. Adsorbate molecules are defined by their SMILES, so they are not uniquely determined by the atom type designation. For example, AdsorbDiff starts from an existing adsorbate configuration (even though torsions are disregarded), which preserves the original bonding. Is bonding information considered in the pipeline? If not, could the authors evaluate whether bonds are reproduced correctly when the generation is adsorbate conditioned?
4. Relaxation of the integer supercell transform is claimed to be "physically meaningful" (Section 3.2), but in what sense is a non integer $M$ a physical object in the structural space? I think unless the non-integer part is interpreted as a strain/shear applied to the lattice, it does not correspond to a periodic replication of the atomic basis. While this + rounding up could still be a practical choice for generative modeling, if the authors can develop and show that canonicalized discrete $M$ generation yields comparable/better results, it could be a nice advancement in generative modeling.

**Limitations:**

Apart from the limitations mentioned in the weaknesses/questions section, the authors have adequately discussed other potential limitations and negative impacts.

**Strengths And Weaknesses:**

### Strengths

- This work falls within catalyst modeling/design, an important domain that has been extensively studied since the early days of ML for science.
- The overall generative pipeline consists of a technically sensible combination of degrees of freedom used in adslab configuration modeling in the domain perspective.
	- One may argue that the combination of an idealized catalyst lattice (without relaxation) and a relaxed adsorbate could be a mismatched modeling choice, but in terms of initial PoC and also considering integration into a full pipeline with downstream energy relaxation, I agree that it would be a reasonable choice for the generative part.
- The exposition in the Method section is structured and written clearly, and the component selections and their rationale are easy to understand.

### Weaknesses

1. Problem setting
	- The biggest concern I have is the problem formulation in this work itself. Catalysis is important, but that does not make a generative model for the OC20 data distribution automatically important. The scientific bottleneck is finding low-energy adslab configurations because eventually adsorption energies matter for catalyst screening, and the standard workflows still enumerate many candidate placements.
	- Since the paper mentions protein-ligand co-generation models in Section 2.3, I would like to illustrate the difference here. PDB structures are experimentally determined structures, which, in theory, are representative of the (free) energy minima of the underlying thermodynamic ensemble. In this way, these models could be applied to generating a meaningful protein binder to the target ligand. However, OC20 is instead a synthetic DFT dataset of surfaces and adsorbates, where placements are chosen by a heuristic generation pipeline. While this is still valuable for MLIP training where we want to sample possible combinations of interactions, this makes distribution learning on OC20 much less inherently meaningful than distribution learning on PDB.
	- More, standard OC20 has only one adsorbate placement per surface, so it is unlikely to be the global minimum for the chosen adsorbate and surface, which is why OC20-dense was introduced in the first place and has been used for previous studies (AdsorbML and AdsorbDiff). Hence, in its current form, it risks appearing as a sophisticated way to imitate a synthetic benchmark rather than a method that materially advances catalyst discovery.
2. Physical stability assessment
	- While the paper assesses the generated structures in terms of validity, uniqueness, and compositional diversity, the current geometric validation does not assess whether the generated catalyst structure is physically meaningful. The established way to assess this in crystalline materials generation literature in a de novo setting is to use SUN/mSUN metrics evaluated using energy over the convex hull. Hence, the validity of structures might have been overestimated beyond what is practically acceptable for a materials generation model.
	- For example, when we look at Figure 4, catalyst structures/compositions in structure prediction (panel b) look reasonable, but some in de novo generation (panel a) look unlikely as a catalyst. For example, In-Rh-Hf ternaries might be fine, but the 9:23:8 stoichiometry and the ordering shown in the structure look quite unlikely. Also, Na3As as a composition exists, but the use of alkali metal pnictide as a catalyst is unlikely, and the generated structure has a weird surface excess of As.
3. Originality claim
	- While the claimed problem of adslab generation and the combination of modeling choices are "new", I do not think they represent significant progress in advancing ML algorithms. While most of the generative modeling comes from existing literature, the additional parts that are unique to this work may be marginal to catalyst modeling.
	- For example, a combination of discrete + continuous flows is adopted from OMatG and has appeared in the literature since early 2024, and lattice parameter sampling is taken from FlowMM. New generative components are the adsorbate, the transformation matrix, and the vacuum scaling factor. However, adsorbate is basically treated similarly but with preconditioned atom types, and transformation matrix and vacuum scaling are just treated as extra continuous dimensions. Also, I want to note that the distribution of supercell and vacuum scaling is dataset-specific: the choices are practically less important to modeling (except for the slab orientation) as long as they are sufficiently large to remove self-interaction or relaxation artifacts.

---

> ### Author Rebuttal · Authors · 2026-03-31
>
> ### **W1, Q1: Unclear problem formulation. OC20 captures only local minima with one placement per surface.**
>
> Thank you for your rigorous and insightful review. We address this point for de novo generation and structure prediction in turn.
>
> ### **De novo generation**
>
> **The goal of our de novo generation is to propose novel compositions worth investigating, not to solve the binding problem in one shot.** A single physically plausible binding configuration suffices as evidence that a proposed surface interacts with the target adsorbate at reasonable energy. Finding the global minimum is a separate downstream step. **[Figure 1](https://imgur.com/TtWXXsB)** illustrates this two-phase pipeline.
>
> **Finetuning on OC20-Dense degrades performance.** As shown in **[Table 1](https://imgur.com/cgxbmF7)**, OC20-Dense covers only ~1K unique systems versus ~460K in OC20. OC20's broad coverage is essential for de novo generation, while OC20-Dense is complementary. We will incorporate this analysis in the revision.
>
> ### **Structure prediction**
>
> **Screening requires a multi-start local search regardless of whether samples come from models trained on OC20 or OC20-Dense**, as illustrated in **[Figure 2](https://imgur.com/P7FQuNA)**. A generative model's value is sample efficiency, i.e., producing candidates closer to good energy basins with fewer samples. We therefore train on OC20 for its broader coverage of diverse slab-adsorbate configurations (compared to OC20-Dense).
>
> **Verification of coverage and OC20-Dense finetuning.**
> We verify that our model trained on OC20 covers the global minima, based on experiments on OC20-Dense, where global minima are known.
>
> In **[Table 2](https://imgur.com/SKLd3je)**, OC20 pretraining alone significantly outperforms random heuristic placement, and OC20-Dense finetuning reduces the success rate for global minima coverage. This confirms that OC20 training captures meaningful thermodynamic interactions, and the slight drop of success rate after OC20-Dense finetuning is consistent with the limited coverage discussed above. We will unify this with the existing structure prediction table in the revision.
>
> ### **W2, Q2: SUN/mSUN metrics for physical stability assessment.**
>
> To incorporate your comment, we evaluate our model with SUN/mSUN metrics in **[Table 3](https://imgur.com/tDroFLe)**. The SUN and mSUN metrics (0.05, 0.82) are on par with ADiT (0.0, 1.0) and SymmCD (0.1, 2.4) evaluated on LeMat-GenBench [3] (bulk crystal generative models trained on MP-20). Low novelty is expected and not necessarily a limitation: CatFlow is trained on OC20 bulk compositions, so it effectively identifies existing stable materials as catalyst candidates, serving a role closer to intelligent screening than de novo materials invention for this case. Pretraining on MP-20 or MPTS-52 could further improve novelty, which we leave as future work.
>
> ### **W3: No unique ML contribution. Core components come from existing work, and new components are just extra continuous dimensions.**
>
> The primary contribution is the problem formulation and representation design, not novel ML components. Prior methods address only isolated pipeline stages, and our framework is the first to formulate slab-adsorbate co-generation as a unified problem.
>
> We also note that the factorized representation is itself an ML contribution. The ablation in **[Table 4](https://imgur.com/LrutsTb)** shows that CatFlow reduces $\Delta E_{sys}$ from 46.8 to 28.0 eV compared to full-slab generation, a gap that would not arise from trivial dimensional augmentation.
>
> ### **Q3: Is bonding information considered in the pipeline?**
>
> As described in Section 3.3, CatFlow does not condition on atomic species alone. We additionally embed reference atomic coordinates based on molecular connectivity and covalent radii, enabling distinction between isomers.
>
> ### **Q4-1: Non-integer $M$ has no physical meaning as a periodic structure unless interpreted as strain/shear on the lattice.**
>
> We agree the phrasing is misleading. Our intention was that the surface orientation is well-defined for any real-valued $M$, not that it produces a valid periodic replication. We will clarify this in the revision.
>
> ### **Q4-2: Discrete generation of $M$ as a potential advancement.**
>
> We replace the continuous relaxation of $M$ with discrete flow matching. **[Table 5](https://imgur.com/jDIaMFi)** shows that this replacement degrades validity, uniqueness, and diversity. Most invalid cases are caused by singular transformation matrices, confirming that continuous relaxation helps generate valid structures. Given the significant validity gap, we prioritize structural metrics, as energy-based metrics are time-consuming and less meaningful when comparing models with very different validity rates.
>
> [1] Zeni et al., Nature 2025
>
> [2] Höllmer et al., ICML 2025
>
> [3] Duval et al., NeurIPS 2025 Workshop

---

> > ### Author Rebuttal · Reviewer_r5u6 · 2026-03-31
> >
> > I appreciate the authors’ thorough response and their effort in conducting extensive evaluations. My concerns have been addressed, and I will increase my score accordingly.

---

> > > ### Author Response · Authors · 2026-04-02
> > >
> > > Thank you for your thoughtful engagement and for reconsidering your evaluation. We are glad our response addressed your concerns.

---

### Official Review · Reviewer_85aP · 2026-03-12

**Soundness:** 3
**Presentation:** 3
**Significance:** 3
**Originality:** 3
**Overall Recommendation:** 5
**Confidence:** 3

**Summary:**

This paper introduces CATFLOW, a flow matching framework for jointly generating slab structures and adsorbate configurations for heterogeneous catalyst design. The key idea is a factorized representation that decomposes the slab-adsorbate system into four components: a primitive cell (atomic species, coordinates, lattice parameters), a transformation matrix (mapping primitive cell to slab supercell), a vacuum scaling factor, and adsorbate coordinates. This factorization reduces the number of learnable variables by an average of 9.2× compared to modeling the full slab directly, since the slab is constructed by replicating the primitive cell. The model uses continuous flow matching for geometric variables (coordinates, lattice parameters, transformation matrix, vacuum factor) and discrete flow matching with masking for atomic species, trained jointly via a single neural network with a transformer-based architecture. Two tasks are addressed: de novo generation (generating both composition and geometry conditioned on adsorbate species) and structure prediction (predicting geometry given known composition). On the OC20 dataset, CATFLOW outperforms CatGPT on de novo generation (97.3% vs. 92.7% structural validity, better uniqueness, lower relaxation energy, higher convergence rate) and substantially outperforms a DiffCSP+AdsorbDiff two-step pipeline on structure prediction (98.2% vs. 65.0% validity, 11.1% vs. 0.01% match rate, 9.7% vs. 1.9% adsorption energy success rate).

**Compliance With Llm Reviewing Policy:**

Affirmed.

**Final Justification:**

Well-prepared rebuttal that substantively addresses every concern with new experiments (four ablation tables) and clear reasoning. My concerns were properly addressed.

**Key Questions For Authors:**

1. **Ablation of the factorized representation.** What happens if you model the full slab directly (without factorization) using the same flow matching framework? This would isolate the contribution of the factorized representation versus the generative model. Even an experiment on a subset of the data would be informative.

2. **Transformation matrix rounding.** What fraction of generated structures have their transformation matrix rounded to the correct integer values? How does rounding error correlate with structural validity and match rate? Is there a failure mode where rounding produces a fundamentally different surface orientation?

3. **Sequential vs. joint generation.** What is the performance if you generate the slab first and then condition adsorbate placement on the generated slab (sequential), using the same factorized representation and flow matching? This would quantify the specific benefit of co-generation versus the representation.

4. **Mixed training data (initial slabs, relaxed adsorbates).** Have you investigated the effect of this mismatch? For instance, does training on both initial and relaxed slab geometries (where decomposable) change performance? Does the model learn to generate slabs that are closer to relaxed geometries despite only seeing initial ones?

**Limitations:**

- The evaluation relies entirely on ML potentials; no DFT validation is provided.
- The factorized representation assumes slabs can be cleanly decomposed into primitive cells, which may not hold for defective surfaces, reconstructed surfaces, or high-entropy alloys.
- Adsorbate conditioning on atomic species alone cannot distinguish structural isomers.
- The framework currently handles single adsorbates; extension to multi-adsorbate systems and coverage effects is mentioned in the conclusion but not explored.
- Computational cost and scaling behavior are not reported — how does generation time compare to baselines and to the DFT cost being replaced?

**Strengths And Weaknesses:**

### Strengths

The factorized representation is the paper's strongest and most original contribution. Decomposing slab-adsorbate systems into primitive cell, transformation matrix, vacuum factor, and adsorbate is physically well-motivated and elegant. The 9.2× average reduction in learnable variables (Figure 2) is substantial and directly addresses a real scalability bottleneck — OC20 slabs average 73 atoms while primitive cells average 18. Importantly, the factorization preserves surface orientation information via the transformation matrix, which is essential for interpreting catalytic activity. This representation could be valuable to the community independently of the specific generative model used.

The co-generation formulation is a meaningful advance over prior work. Table 1 clearly positions CATFLOW relative to competitors: Catalyst GFlowNet generates bulk but not slabs, AdsorbDiff places adsorbates but requires fixed slabs, and CatGPT generates sequentially without modeling surface-adsorbate coupling. CATFLOW is the first to jointly generate all components within a unified objective, which is conceptually the right approach since adsorbate binding depends on surface geometry and vice versa.

The experimental design for de novo generation is thorough. Beyond standard validity and uniqueness, the authors evaluate relaxation statistics (energy change, convergence steps, convergence rate) which provide meaningful evidence that generated structures are near local minima. The adsorption energy distribution analysis (Figure 5) across 14 adsorbates with KDE plots comparing to validation set distributions is informative and shows that CATFLOW produces more physically realistic energy landscapes than CatGPT, which tends to generate unstable configurations with positive-energy tails.

The structure prediction comparison is set up in a way that actually disadvantages CATFLOW: the DiffCSP+AdsorbDiff baseline receives ground-truth transformation matrices and vacuum scaling factors, while CATFLOW predicts them from scratch. Despite this handicap, CATFLOW substantially outperforms the baseline, which is a strong result.

The paper is clearly written, with good figures (especially Figures 1, 3, and 4) that help communicate the factorized representation and generation process. The data processing pipeline (Appendix B) is described in detail, and code and data are provided via anonymous links.

### Weaknesses

**Absolute performance on structure prediction remains low.** While CATFLOW clearly outperforms the baseline, the absolute numbers for structure prediction are concerning: 11.09% match rate and 9.72% adsorption energy success rate (within 0.1 eV). These are far from practically useful levels. The authors acknowledge that conditioning on composition inherently yields diverse valid slabs, which affects match rates, but this limitation deserves more discussion. Is structure prediction even the right framing if the task admits many valid solutions for a given composition? The paper would benefit from discussing whether generating multiple valid structures (rather than matching a single ground truth) is a more appropriate evaluation paradigm.

**Limited baseline comparison for de novo generation.** CatGPT is the only baseline for de novo generation. While the authors argue (correctly) that no other method satisfies all the criteria in Table 1, the improvements over CatGPT are modest on some metrics — both achieve >92% validity and >15 compositional diversity. Stronger baselines or ablations would help quantify the benefit of co-generation versus other design choices (e.g., the factorized representation alone, the flow matching formulation alone). The paper lacks an ablation study isolating the contributions of individual components.

**No ablation study.** This is a significant gap. The paper combines several innovations — factorized representation, co-generation, continuous+discrete flow matching, the specific transformer architecture — but never isolates their individual contributions. Key questions remain unanswered: How much does the factorized representation help versus modeling the full slab directly? How much does co-generation (joint slab+adsorbate) help versus sequential generation? What is the effect of the transformation matrix relaxation from integers to continuous values during training? An ablation table decomposing these effects would substantially strengthen the paper.

**Transformation matrix relaxation.** The paper relaxes the integer-valued transformation matrix M to continuous values during training and rounds at inference. This is an important design choice that is insufficiently analyzed. How often does rounding produce the correct M? What is the distribution of rounding errors? When rounding fails, what types of structural errors result? The paper states that "a real-valued M still defines a valid surface orientation" but this is not quite right — the resulting slab may not tile correctly if M is not integer. Some analysis of the sensitivity to rounding errors would be valuable.

**Training on initial slabs but relaxed adsorbates.** The paper uses initial (unrelaxed) slab structures for extracting the primitive cell, transformation matrix, and vacuum factor, but uses relaxed adsorbate coordinates. This mixed strategy is motivated by the fact that relaxed surfaces break primitive cell periodicity, but it introduces an inconsistency: the slab geometry does not correspond to the energy minimum, while the adsorbate placement does. The implications of this mismatch for the learned distribution are not discussed. Does the model learn to generate slab structures that are physically meaningful despite never seeing relaxed slab geometries?

**Evaluation relies entirely on ML potential, not DFT.** All relaxations use the uma-s-1p1 model rather than DFT. While this is practical, the paper does not validate that the ML potential accurately captures the relevant physics for the generated structures, which may be out-of-distribution relative to the potential's training data. Even a small DFT verification on a subset of generated structures (as done in some related work) would increase confidence.

**Limited analysis of failure modes.** The paper does not analyze what goes wrong in the ~3% of invalid de novo generations or the ~2% of invalid structure predictions. Understanding failure modes (e.g., unrealistic coordination, broken periodicity, atom overlaps) would provide useful diagnostics.

**Adsorbate conditioning limited to atomic species.** The framework conditions on adsorbate atomic species but not on the adsorbate's molecular graph or connectivity. For adsorbates with the same atomic composition but different connectivity (e.g., structural isomers), the model cannot distinguish between them. This limitation is not discussed.

---

> ### Author Rebuttal · Authors · 2026-03-31
>
> Thank you for your detailed review. We respond point-by-point.
>
> ### **W1: Low absolute structure prediction performance and whether single ground-truth matching is the right evaluation framing.**
>
> A given composition can admit multiple valid slabs, so low match rate does not imply poor practical utility. In a screening scenario, the goal is to explore diverse configurations that reach physically meaningful local minima, not to recover a single ground-truth. While we adopt match rate and RMSD following CSP benchmarks [1], we additionally report the $\Delta E_{ads}$ success rate to evaluate whether the generated structures are energetically plausible. We note that match rate still serves as a useful lower bound on model quality.
>
> ### **W2, W3, Q1: Lack of ablation studies.**
>
> We ablate CatFlow against a full-slab generation variant in **[Table 1](https://imgur.com/X4M3FYy)** and a sequential generation variant in **[Table 2](https://imgur.com/R09vDGF)**, all trained for the same duration. All variants are evaluated on the de novo generation task. Both design choices contribute to CatFlow's superior performance.
>
> **[Table 3](https://imgur.com/ySoeT4C)** and **[Table 4](https://imgur.com/fk1gmTb)** ablate architecture and continuous relaxation of $M$, respectively. Both replacements degrade structural validity. For discrete training of $M$, the majority of invalid cases are caused by singular transformation matrices, confirming that the continuous relaxation helps generate valid structures. Given the significant validity gap, we prioritize structural metrics, as energy-based metrics are time-consuming and less meaningful when comparing models with very different validity rates, and so is uniqueness.
>
> ### **W4-1,Q2: Missing analysis for rounding accuracy of $M$.**
>
> We respectfully argue that analyzing the rounding accuracy of $M$ in isolation is not meaningful. $M$ is jointly generated with the primitive cell, and even when neither exactly matches the ground truth, their combination can still produce a valid slab. In de novo generation, there is no ground-truth $M$. In structure prediction, multiple $M$ can yield valid slabs, so requiring an exact match would penalize diversity.
>
> ### **W4-2: Wrong statement for real-valued $M$. The resulting slab may not tile correctly.**
>
> We agree the phrasing is misleading and will clarify that the surface orientation is well-defined for any real-valued $M$, not that it produces a valid periodic replication of the primitive cell.
>
> ### **W5,Q4: Does the mixed structure correspond to a physically meaningful energy state?**
>
> Our design treats the slab as a fixed host and learns the energetically favorable binding geometry of the adsorbate on it. The generated structure is not the final prediction but an initialization for downstream relaxation, and the essential point is that it starts close enough to a good energy basin for relaxation to converge. Our results support this choice, as generated structures reliably converge to local minima during MLIP relaxation. We note that this fixed-host assumption is widely adopted in related domains of protein-ligand docking, where the protein structure is typically held rigid while only the ligand pose is predicted, with full relaxation deferred to a subsequent step.
>
> ### **W6, L1: Lack of DFT evaluation.**
>
> DFT evaluation for all generated samples is prohibitively expensive. To ensure a fair comparison, we evaluate the reference, our framework, and all baselines using the same UMA model, which is known to achieve DFT-level accuracy for catalytic applications.
>
> ### **W7: Analysis of structural invalid cases.**
>
> In both tasks, most invalid cases (97.74%) are caused by atom overlap, with the remainder caused by singular transformation matrices.
>
> ### **W8, L3: Only atomic species of adsorbate cannot distinguish isomers.**
>
> As described in Section 3.3, CatFlow conditions not only on atomic species but also on reference atomic coordinates from molecular connectivity and covalent radii, which enables isomer distinction.
>
> ### **Q3: How about generating the slab first and then adsorbate placement sequentially?**
>
> Our DC+AD baseline is exactly this. It follows the same factorized representation and sequential generation by generating the primitive cell (DiffCSP), combining it with ground truth transformation matrix and vacuum scaling factor, and then places the adsorbate (AdsorbDiff).
>
> ### **L2, L4: Defective/reconstructed surfaces, high-entropy alloys, and multi-adsorbate systems.**
>
> These extensions require dedicated datasets and modeling beyond the current scope. We consider them promising future directions.
>
> ### **L5: Computational cost and scaling behavior are not reported.**
>
> CatFlow achieves an inference throughput of 1.041 samples/sec, outperforming CatGPT (0.325) and DC+AD (0.046). Combined with MLIP relaxation, this pipeline is orders of magnitude faster than full DFT optimization.
>
> [1] Höllmer et al., ICML 2025

---

> > ### Author Rebuttal · Reviewer_85aP · 2026-04-02
> >
> > Well-prepared rebuttal that substantively addresses every concern with new experiments (four ablation tables) and clear reasoning. I will increase my score accordingly.

---

> > > ### Author Response · Authors · 2026-04-03
> > >
> > > Thank you for your thorough review and generous assessment of our rebuttal. We are glad the additional experiments and clarifications were convincing.

---

### Official Review · Reviewer_gA6r · 2026-03-13

**Soundness:** 3
**Presentation:** 3
**Significance:** 3
**Originality:** 3
**Overall Recommendation:** 5
**Confidence:** 2

**Summary:**

This paper presents CATFLOW, a flow matching–based framework for de novo design and structure prediction of heterogeneous catalysts. The framework jointly co-generating slab structures and adsorbate coordinates using a primitive cell–based factorized representation. Experiments on the OC20 show that CATFLOW improves the structural fidelity of generated catalysts and produces structures closer to thermodynamic local minima compared.

**Compliance With Llm Reviewing Policy:**

Affirmed.

**Final Justification:**

My concerns have been addressed. Thus, I have increased my score.

**Key Questions For Authors:**

1. Why is only OC20 used but not OC22 or OC25 for the experiments?
2. What is the computational cost of CatFlow compared to other baselines for training and inference?

**Limitations:**

Yes.

**Strengths And Weaknesses:**

**Strength**
1. CATFLOW is a novel framework to co-generate slab structures and adsorbate coordinates within a unified flow matching objective, achieving superior performance compared to previous models.
2. The factorized representation significantly reduces learnable variables while preserving surface information critical for catalytic systems.


**Weakness**
1. Even though CatFlow outperforms the baselines, the match rate and success rate are still relatively low at only around 10%.
2. Figure 5 only provides KDE plots, but the author could report quantitative metrics to substantiate the claim that CatFlow better matches reference energy distributions.
3. The lack of ablations makes it unclear whether the gains come from the factorized representation, the co-generation objective, or other design choices.

---

> ### Author Rebuttal · Authors · 2026-03-31
>
> Thank you for your review. We address your concerns regarding ablations, metrics, and experiments in detail.
>
> ### **W1: Even though CatFlow outperforms, the match rate and $\Delta E_{ads}$ success rate in structure prediction task are low.**
>
> The low absolute numbers are expected because conditioning on composition alone admits multiple valid slab structures, each with different adsorption energies. Both MR and success rate are evaluated against a single ground truth, penalizing valid but structurally distinct predictions. Nevertheless, match rate serves as a useful lower bound on model quality, and CatFlow consistently outperforms baselines under this metric.
>
> ### **W2: There is no quantitative metric for how well CatFlow matches the reference $\Delta E_{ads}$ distributions.**
>
> We evaluate the 2-Wasserstein distance between the generated and reference $\Delta E_{ads}$ distributions per adsorbate in the validation set. CatFlow achieves an average 2-Wasserstein distance of 0.603, outperforming CatGPT (0.766).
>
> ### **W3: There is no ablation for factorized representation, co-generation objective, or other design choices.**
>
> **Ablation for factorized representation and co-generation.** We compare CatFlow against a full-slab generation variant (**[Table 1](https://anonymous.4open.science/r/anonymous_rebuttal-7FD0/2_gA6r/Table1.png)**) and a sequential generation variant (**[Table 2](https://anonymous.4open.science/r/anonymous_rebuttal-7FD0/2_gA6r/Table2.png)**), all trained for the same duration.
> All variants are evaluated on the de novo generation task. The results confirm that both design choices contribute to CatFlow's superior performance.
>
> **Ablation for architectures and continuous relaxation of $M$.** We conduct ablation studies for the architecture (**[Table 3](https://anonymous.4open.science/r/anonymous_rebuttal-7FD0/2_gA6r/Table3.png)**) and continuous relaxation of $M$ (**[Table 4](https://anonymous.4open.science/r/anonymous_rebuttal-7FD0/2_gA6r/Table4.png)**). The results show that both replacements degrade structural validity. For discrete training of $M$, the majority of invalid cases are caused by singular transformation matrices, confirming that the continuous relaxation helps to generate valid structures. Given the significant validity gap, we prioritize structural metrics, as energy-based metrics are time-consuming and less meaningful when comparing models with very different validity rates. Uniqueness is also less interpretable under such conditions.
>
>
> ### **Q1: Why is only OC20 used but not OC22 or OC25 for the experiments?**
>
> We use OC20 as the first benchmark for slab-adsorbate co-generation because it is the largest and most widely used dataset for heterogeneous catalysis. OC22 and OC25 introduce additional complexity such as oxide surfaces and explicit solvent environments, which we consider natural extensions of CatFlow in future work.
>
> ### **Q2: What is the computational cost of CatFlow compared to other baselines for training and inference?**
>
> We provide the training and inference time for all methods in **[Table 5](https://anonymous.4open.science/r/anonymous_rebuttal-7FD0/2_gA6r/Table5.png)**. For inference, CatFlow achieves the highest throughput at 1.041 samples/sec, compared to CatGPT (0.325) and DC+AD (0.046).

---

> > ### Author Rebuttal · Reviewer_gA6r · 2026-04-03
> >
> > Thank you for the response. My concerns have been addressed. I have updated the score.

---

> > > ### Author Response · Authors · 2026-04-03
> > >
> > > Thank you for your careful review and for updating your score. We are pleased our response resolved your concerns.

---

### Official Review · Reviewer_PgSb · 2026-03-18

**Soundness:** 3
**Presentation:** 3
**Significance:** 2
**Originality:** 3
**Overall Recommendation:** 4
**Confidence:** 4

**Summary:**

CatFlow is a flow matching framework applied for the co-generation of adsorbates and slabs. The work also contributes a novel factorized representation for slab-adsorbate systems, which helps in reducing the dimensionality of the overall problem. Since the overall approach of denovo generation (with structure) is novel and there is no fair prior art for the work, the authors test this against the baseline of a combination of DiffCSP and AdsorbDiff framework.

**Compliance With Llm Reviewing Policy:**

Affirmed.

**Key Questions For Authors:**

(many questions are already described in weaknesses)
- What do you think are the reasons for the significant jump in Uniqueness for de novo generation using CatFlow?
- Is the success % low for structure prediction task primarily due to invalid generations from DiffCSP?

**Limitations:**

yes

**Strengths And Weaknesses:**

Strengths:

- This is a novel framework that advances the field of de novo generation of adsorbate and slab systems. None of the prior work in the field, like AdsorbDiff and CatGPT, has specifically focused on this.
- This approach demonstrates superior results in comparison to CatGPT for de novo generation task and DC + AD baseline (more questions on this later).

Weaknesses:

- Several key ablations are missing to understand the impact of each variable that's changed over the baseline
  - AdosrbDiff uses a GemNet/ Equiformer V2 / PaiNN as optimizers, whereas this work uses uma models. Its unclear, how well the flow matching does in this setup.
  - The impact of change in architecture is unclear as well.
  - Is it possible to run an ablation to show the impact of factorized representation?
  - A fair comparison could be finetuning DiffCSP on the bulks of those catalyst surfaces for the structure generation task?
 - Although I understand the importance of de novo generation of catalysts, it's unclear to me if the data that this is being trained on is any meaningful, as this was not the purpose of generating the data.
    - To elaborate, the data doesn't necessarily represent a minima across all possible catalysts. The goal of adsorbML work was to generate energetics of all possible adsorbate configurations given a catalyst, but it was not necessarily looking at the best catalyst surface for a given adsorbate exhaustively. This is certainly an interesting problem to solve, but its unclear to me if the metrics calculated in this work truly demonstrate improvement in this direction (happy to be corrected on this if I'm misinterpreting anything).

Overall, it seems there are many changes relative to the baselines this work compares against, and it is unclear which changes lead to the overall improvements.

---

> ### Author Rebuttal · Authors · 2026-03-31
>
> Thank you for your review. We address your concerns regarding ablations and data in detail.
>
> ### **W1: Missing ablation studies**
>
> We provide the requested ablation studies below.
>
> #### **W1-1: Different MLFF models.**
>
> We evaluate the $\Delta E_{ads}$ success rate (tolerance $\leq$ 0.1 eV) using the same GemNet-OC as AdsorbDiff. **[Table 1](https://anonymous.4open.science/r/anonymous_rebuttal-7FD0/1_PgSb/Table1.png)** shows that CatFlow outperforms the baseline regardless of the relaxation model.
>
> #### **W1-2: Different architectures.**
>
> We compare our DiT with EGNN for de novo generation. **[Table 2](https://anonymous.4open.science/r/anonymous_rebuttal-7FD0/1_PgSb/Table2.png)** shows that EGNN achieves only 49.68% structural validity, confirming that architecture choice significantly affects performance. Given this large validity gap, we prioritize structural metrics, as energy-based metrics are time-consuming and less meaningful when comparing models with very different validity rates. Uniqueness is also less interpretable under such conditions.
>
> #### **W1-3: Necessity of factorized representation**
>
> We compare with a full-slab generation variant. The results in **[Table 3](https://anonymous.4open.science/r/anonymous_rebuttal-7FD0/1_PgSb/Table3.png)** show that the factorized representation captures physically plausible slab-adsorbate structures more effectively, as measured by adsorption energy.
>
> #### **W1-4: Finetuning DiffCSP on the OC20 bulks**
>
> We already trained DiffCSP on the primitive cell structures in OC20 for a fair comparison. We will clarify this in the revised paper.
>
> ### **W2: Suitability of OC20 for de novo catalyst generation and whether metrics demonstrate meaningful improvement.**
>
> We appreciate the reviewer raising this important point. CatFlow's goal is not to find the single globally optimal surface for a given adsorbate. In practice, surfaces with the most reactive binding sites (especially high Miller indices) tend to have high surface energy and are thermodynamically unstable [1, 2], as illustrated in **[Table 4](https://anonymous.4open.science/r/anonymous_rebuttal-7FD0/1_PgSb/Table4.png)**.
>
> Given this reactivity–stability trade-off, the practically relevant objective is exploring diverse binding configurations across plausible surfaces, then feeding promising candidates into downstream validation. Our de novo generation filters promising catalyst candidates based on "hints" from generated slab-adsorbate configurations, to be processed through additional pipelines. See **[Figure 1](https://anonymous.4open.science/r/anonymous_rebuttal-7FD0/1_PgSb/DNG_pipeline.png)** for the full pipeline.
>
> Under this framing, our metrics are well-suited: uniqueness and compositional diversity measure exploration breadth, while adsorption energy distributions (Figure 5) and relaxation statistics (Table 2) confirm physical plausibility, not global optimality, but thermodynamically reasonable candidates for screening. We will clarify this workflow and its connection to the metrics in the revision.
>
> ### **Q1: Reason for better uniqueness of CatFlow?**
>
> In autoregressive models, each token is sampled conditioned on all previous tokens. Since high-frequency compositions dominate training data, independent samples tend to follow similar high-probability trajectories, producing duplicates. CatGPT acknowledged this and tried increasing temperature, but at the cost of structural validity [4].
>
> In CatFlow, each sample starts from an independent continuous noise vector, so diversity comes from the noise prior rather than data frequency. This decoupling is a known advantage of continuous generative models over autoregressive ones [3].
>
> ### **Q2: Is the low performance of DC+AD primarily caused by DiffCSP generating invalid primitive cells?**
>
> Not primarily. The DC+AD pipeline achieves 64.95% structural validity, but even for valid structures, the transformation matrix $M$ amplifies small errors in the predicted primitive cell during slab construction. AdsorbDiff then receives these imperfect slabs as out-of-distribution inputs. More fundamentally, the two-step pipeline cannot capture the coupling between slab geometry and adsorbate placement, which is the core advantage of CatFlow's end-to-end co-generation.
>
> [1] Tran et al., Scientific Data, 2016
>
> [2] Medford et al., Journal of Catalysis, 2015
>
> [3] Ye et al., NeurIPS 2024
>
> [4] Mok & Back, JACS, 2024

---

> > ### Author Rebuttal · Reviewer_PgSb · 2026-04-01
> >
> > Authors have addressed my concerns and therefore I'm increasing my score.

---

> > > ### Author Response · Authors · 2026-04-02
> > >
> > > Thank you for your engagement and for updating your score. We appreciate your positive feedback on our response.

---

### Decision · Program_Chairs · 2026-04-30

**Decision:**

Accept (regular)

**Comment:**

This paper has received four reviews, all of them of sufficient quality and several of them of more-than-average quality. The authors responded too most concerns and questions from the reviewers and the reviewers have unanimously stated that their concerns were successfully addressed and recommended the acceptance of the paper. Therefore, my decision is, accordingly, to recommend the acceptance of this paper.

The concerns shared by several reviewers were the lack of ablation studies and the low performance in absolute terms. The authors have provided a number of additional ablation studies, which have satisfied the reviewers. I encourage the authors to include these in their updated manuscript. The practically low performance has been explained by the authors and, while it has not been further discussed during the rebuttal, in my opinion it does not undermine the value of the paper. It rather speaks about the difficulty of the task. Other comments from the reviewers are relevant and have initiated interesting answers from the authors which I also encourage to incorporate in the manuscript.

Finally, the reviews do not mention any serious technical or quality issues. All this considered, I think this is a good quality paper, about a relevant topic for the ML for science and materials discovery community.